# Unipolar distributions of junctional Myosin II identify cell stripe boundaries that drive cell intercalation throughout *Drosophila* axis extension

Robert J Tetley[1†], Guy B Blanchard[1*†], Alexander G Fletcher[2,3], Richard J Adams[1], Bénédicte Sanson[1*]

[1]Department of Physiology, Development and Neuroscience, University of Cambridge, Cambridge, United Kingdom; [2]School of Mathematics and Statistics, University of Sheffield, Sheffield, United Kingdom; [3]Bateson Centre, University of Sheffield, Sheffield, United Kingdom

**\*For correspondence:** gb288@ cam.ac.uk (GBB); bs251@cam.ac. uk (BS)

[†]These authors contributed equally to this work

**Competing interests:** The authors declare that no competing interests exist.

**Abstract** Convergence and extension movements elongate tissues during development. *Drosophila* germ-band extension (GBE) is one example, which requires active cell rearrangements driven by Myosin II planar polarisation. Here, we develop novel computational methods to analyse the spatiotemporal dynamics of Myosin II during GBE, at the scale of the tissue. We show that initial Myosin II bipolar cell polarization gives way to unipolar enrichment at parasegmental boundaries and two further boundaries within each parasegment, concomitant with a doubling of cell number as the tissue elongates. These boundaries are the primary sites of cell intercalation, behaving as mechanical barriers and providing a mechanism for how cells remain ordered during GBE. Enrichment at parasegment boundaries during GBE is independent of Wingless signaling, suggesting pair-rule gene control. Our results are consistent with recent work showing that a combinatorial code of Toll-like receptors downstream of pair-rule genes contributes to Myosin II polarization via local cell-cell interactions. We propose an updated cell-cell interaction model for Myosin II polarization that we tested in a vertex-based simulation.

## Introduction

Polarised cell rearrangements drive the simultaneous elongation and narrowing of cell sheets (convergence and extension) during development. These collective cell behaviours have been mostly studied in the context of axis elongation that accompanies gastrulation in bilaterian animals, but are also found in organogenesis, for example underlying kidney tubule elongation (*Keller, 2002*; *Tada and Heisenberg, 2012*). Understanding convergence and extension movements is important as their failure is associated with congenital diseases, including neural tube defects (*Wallingford et al., 2013*). The first molecular mechanism for convergence and extension was found in *Drosophila*, where planar polarisation of actomyosin was shown to underlie the polarised cell rearrangements of germband extension (GBE) (*Zallen and Wieschaus, 2004*; *Bertet et al., 2004*). This discovery paved the way for in-depth studies of how the planar polarisation of actomyosin and other components such as Bazooka (Par-3) and E-Cadherin drives the selective shortening of cell-cell junctions during active intercalation of epithelial cells (*Zallen and Wieschaus, 2004; Rauzi et al., 2008*; *2010*; *Levayer et al., 2011*; *Levayer and Lecuit, 2013*; *Blankenship et al., 2006*; *Fernandez-Gonzalez et al., 2009*; *Simões et al., 2010*; *2014*; *Tamada et al., 2012*). Recently, actomyosin planar polarisation was also found to be required during convergence and extension in vertebrate

**eLife digest** Early in development, a growing embryo elongates to form its main body (head–tail) axis. This elongation is driven by a process called cell intercalation – when cells insert between each other. The mechanism that controls this coordinated cell movement is well understood on a small scale. However, it is not known how hundreds of cells rapidly intercalate across a whole tissue without deforming a tissue or inappropriately mixing.

During fruit fly development, an embryo divides into repeated segments of tissue while elongating. While this happens, cells redistribute an essential structure called the actomyosin cytoskeleton so that it is found more commonly along certain sides of the cell. This structure, which can be thought of as the cell's "muscle", is a contractile web made of proteins called actin and myosin. It is closely associated with the cell's membrane and causes cells to contract and push past each other. The enrichment of the actomyosin cytoskeleton on certain sides of a cell is determined by signaling systems, which are controlled by the segmentation genes in the fruit fly and by the so-called planar cell polarity pathway in vertebrates.

Tetley, Blanchard et al. have now investigated cell intercalation across a whole tissue by filming live fruit fly embryos in which both actomyosin and cell membranes were made visible with fluorescent markers. Computational tools were then used to quantify how much actomyosin is enriched in the sides of thousands of cells in the embryo at particular points in time while the embryos elongated. This revealed reproducible patterns of actomyosin enrichment. As embryos elongated, the actomyosin cytoskeleton redistributed itself inside the cells: whereas at the start two opposite sides of each cell were enriched in actomyosin (a bipolar distribution), at later times the enrichment occurred on just one side (a unipolar distribution).

Incorporating these patterns into a model of tissue-wide cell intercalation showed that cells along the head–tail axis acquire a specific identity depending on their position. Interactions between the cells then allow the cells to compare their identities with each other and modify their pattern of actomyosin enrichment accordingly. Where the identities of neighbouring cells are different, the cells enrich actomyosin along their shared sides, creating boundaries between stripes of cells that share the same identity.

These findings show that actomyosin-rich boundaries drive the elongation of the head–tail axis while limiting cell intermixing. Future work will investigate how the patterns of actomyosin enrichment are altered in fly mutants in which the identities of the cells along the head–tail axis are disrupted.

tissues (*Rozbicki et al., 2015*; *Nishimura et al., 2012*; *Lienkamp et al., 2012*; *Shindo and Wallingford, 2014*).

The upstream signals that pattern these polarities in the plane of the converging and extending tissues are starting to be deciphered. In vertebrates, the conserved planar cell polarity (PCP) pathway controls planar cell rearrangements during axis extension (*Wallingford, 2012*). In the *Xenopus* model, this pathway was recently shown to do so by biasing the polarisation of actomyosin (*Shindo and Wallingford, 2014*). In *Drosophila*, the PCP pathway is not required for polarisation of the actomyosin cytoskeleton (*Zallen and Wieschaus, 2004*), which instead depends on the segmentation cascade, the most downstream cues being the striped expression of pair-rule transcription factors such as Eve or Runt (*Zallen and Wieschaus, 2004*; *Bertet et al., 2004*). Misexpression of these pair-rule transcription factors causes a local reorientation of polarity, which led to the hypothesis that local cell-cell interactions generate planar polarity in the *Drosophila* germband, rather than more global cues (*Zallen and Wieschaus, 2004*). Recent work has provided molecular evidence for this; three Toll-like receptors are expressed in overlapping stripes in the early embryo under the control of the pair-rule genes *eve* and *runt* (*Paré et al., 2014*). Genetic disruption of these receptors leads to defects in GBE and a corresponding loss of the planar polarisation of Myosin II and Bazooka in the tissue. A model was proposed in which the germband is planar polarised through the preferential enrichment of Myosin II at sites of heterophilic Toll-like receptor interactions (*Paré et al., 2014*). The overlapping expression domains of Toll-like receptors would therefore establish a

combinatorial code where every cell along the antero-posterior (AP) axis has a different 'identity', resulting in the bipolar distribution of Myosin II in every cell.

These findings open new questions. One is what becomes of the combinatorial code and the planar polarisation of Myosin II once the cells have started intercalating and the number of cells increases along AP? Specifically, if the cell identity stripes defined by the Toll-like receptor code are one cell wide to start with as hypothesised (*Paré et al., 2014*), then these would increase to two cells wide on average after one round of cell intercalation. Heterophilic interactions between Toll receptors would no longer be expected at the interfaces between pairs of cells of the same 'identity'. Therefore one possibility is that these interfaces are not enriched in Myosin II at later stages of GBE. Alternatively, a secondary mechanism might be required to polarise the germband in later GBE, for example relying on a global polarising signal, more akin to PCP pathway-reliant polarisation in vertebrates (*Devenport, 2014*; *Goodrich and Strutt, 2011*).

Another unsolved question is how the AP patterns established early in development are maintained during the cell movements of convergent extension (*Dahmann et al., 2011*; *Vroomans et al., 2015*). Cell rearrangements by intercalation are sufficient to cause mixing of adjacent cell populations (*Umetsu et al., 2014*), therefore it is likely that a mechanism exists to maintain order along the AP axis of the germband. At later stages of embryonic development in *Drosophila*, an enrichment of actomyosin at parasegmental boundary (PSB) cell-cell interfaces is required to prevent cell intermingling caused by cell proliferation (*Monier et al., 2010*; *2011*). The actomyosin enrichment in this case is thought to act as a mechanical barrier, since the enriched PSB cell-cell interfaces align, indicating line tension. Supporting this notion, laser ablation experiments have demonstrated an increase in interfacial tension at compartmental boundaries in the wing disc and abdomen (*Umetsu et al., 2014*; *Aliee et al., 2012*; *Landsberg et al., 2009*). Since parasegmental boundaries are defined genetically by pair-rule gene expression before gastrulation starts (*Lawrence and Johnston, 1989*), an unexplored possibility is that actomyosin enrichments at PSBs could form early, during GBE, and limit intermingling of cells during the large-scale cell rearrangements of convergence and extension.

Here we take a systems biology approach to answer these questions by investigating the relationship between segmentation, the planar distribution of the motor Myosin II and the cell behaviours contributing to axis extension. We aimed to develop an analysis of these at the scale of the tissue, in living wild-type embryos. In particular, we asked what the relationship is between the described bipolar distribution of Myosin II at AP interfaces early in GBE and the later formation of parasegmental boundaries that stop mixing between anterior and posterior compartments. We show that Myosin II has a bipolar distribution in early embryos, which then transitions to a unipolar distribution as a direct consequence of polarised cell intercalation in the germband. Such an observation strongly supports that a cell identity mechanism polarises Myosin II throughout the whole of GBE. We show that the boundaries defined by the unipolar patterns, which include the PSBs, are the sites of the cell intercalation events driving GBE. We demonstrate that the PSB is a distinct mechanical structure from very early in GBE. These findings suggest that the boundaries we identify have a dual role, driving axis extension while ensuring that cell mixing remains limited. Finally, we propose an updated differential cell identity model.

## Results

### Bidirectional polarity of Myosin II is short-lived during axis extension and unidirectional polarity patterns soon dominate

We reasoned that analysing the spatiotemporal modulations of actomyosin enrichment during GBE might answer the above questions by revealing undiscovered patterns. To quantify changes in Myosin II polarisation during GBE, we imaged the ventral surface of *Drosophila* embryos co-expressing the fluorescent fusion proteins *GAP43-mCherry* (*Martin et al., 2010*), to label the cell membranes, and *sqh-GFP* (*Royou et al., 2004*), to label Myosin II (*Figure 1A*, *Video 1*). Because *sqh-GFP* was expressed in a *sqh^AX3* null mutant background, all Myosin II molecules were tagged with GFP (*Royou et al., 2004*). Images were acquired every 30 s, from before the start of extension, until the enrichment of Myosin II at parasegmental boundaries (PSBs) (*Monier et al., 2010*) was clearly detectable at the end of extension (*Video 1*, *Figure 2A*). The *GAP43-mCherry* signal was used to

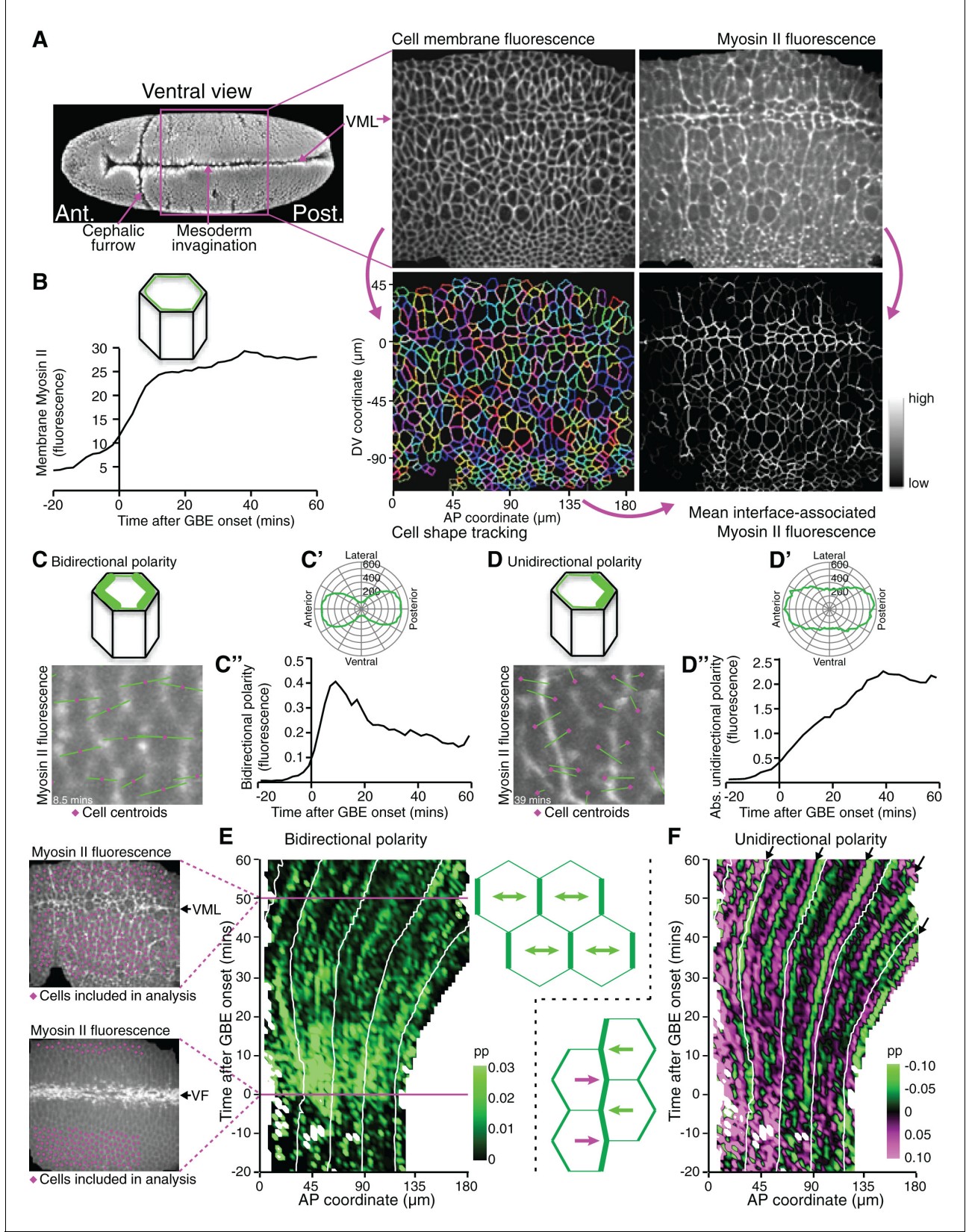

**Figure 1.** Quantifying Myosin II polarisation over time during *Drosophila* axis extension. (A) *sqh^AX3*; *sqh-GFP*; *GAP43-mCherry* embryos are imaged ventrally by confocal microscopy with a 196 x 173 μm field of view, with cell membranes visualised in the red channel and Myosin II in the green

*Figure 1 continued on next page*

*Figure 1 continued*

channel. Apical cell-cell interfaces are tracked over time based on the cell membrane signal. Next, Myosin II fluorescence intensities associated with the tracked cell-cell interfaces are quantified. Six movies were collected. VML: ventral midline. SEM image on the left from Flybase (*dos Santos et al., 2015*). (B) Total fluorescence intensities for Myosin II at apical cell-cell interfaces over time. Data shown in B, C', C", D' and D" is averaged for the 6 movies. (C–C") Quantification of Myosin II bidirectional polarity. (C) Fourier quantification of Myosin II bipolarity, depicted here on a movie frame 8.5 min after GBE onset. The length of the bipolar green vector represents the amplitude of polarity and its angle, the orientation of the polarity relative to the AP embryonic axis. Because the polarity is essentially aligned along the AP embryonic axis (rosette in C'), the polarity amplitude can be projected onto the AP axis and quantified using a Gaussian fit which allows a better separation between bidirectional and unidirectional polarity signals, compared to the Fourier method (*Figure 1—figure supplement 2*). (C") Amplitude of Myosin II bidirectional polarity along the AP axis and over time, calculated using the Gaussian method. (D) Fourier quantification of Myosin II unidirectional polarity, shown on a movie frame 39 min after GBE onset. The length of the unipolar green vector represents the amplitude of polarity and its slope, the orientation of the polarity relative to the AP embryonic axis (see corresponding rosette in D'). The vector either points towards the anterior or the posterior, depending which side of a given cell is enriched in Myosin II. (D") Absolute amplitude of Myosin II unidirectional polarity along the AP axis and over time, calculated using the Gaussian method (*Figure 1—figure supplement 2*). (E) Spatio-temporal map showing Myosin II bidirectional polarity for one representative movie (SG_4, *Figure 1—figure supplement 1* and *3*), as a function of the AP position in the field of view (x-axis, in µm) and time relative to the start of GBE (y-axis, in min). Movie frames corresponding to 0 and 50 min are shown on the left. At time 0, the mesoderm is invaginating through the ventral furrow (VF, white streak in the middle of the image). Mesoderm and mesectoderm cells on either side of the VF are not included in the analysis, nor are the germband cells at the posterior, because these move out of the field of view with the convergence extension of the tissue. Germband cells included in the analysis are labelled in magenta on both frames. At 50 min, most of the cells in the field of view are included in the analysis, except the mesectoderm cells at the midline (VML) and very dorsal germband cells coming in the field of view (bottom). The amplitude of Myosin II bipolarity is expressed as a proportion (Abbreviated as pp in all figures) of the mean Myosin II intensity around the perimeter of each cell. Scale shows highest bidirectional polarity in bright green and no polarity in black. White lines on the plot follow the displacement of AP coordinates over time, which move posteriorly as the tissue undergoes extension. (F) Spatio-temporal map showing Myosin II unidirectional polarity for the same representative movie. The amplitude of unipolarity is expressed as a proportion (pp) of mean Myosin II intensities at cell-cell interfaces. Scale shows enrichment towards anterior cell-cell interfaces as green (negative values) and towards posterior as magenta (positive values). Input data and statistics are in *Figure 1—source data 1*.

The following source data and figure supplements are available for figure 1:

**Source data 1.** Source data for *Figure 1*, including statistical analysis.
**Source data 2.** Source data for *Figure 1—figure supplement 1*, including statistical analysis.
**Figure supplement 1.** Synchronisation of *sqh^{AX3}; sqh-GFP; GAP43-mCherry* movies.
**Figure supplement 2.** Methods for calculating bidirectional and unidirectional Myosin II polarity.
**Figure supplement 3.** Spatiotemporal maps for all *sqh^{AX3}; sqh-GFP; GAP43-mCherry* movies.

segment and track apical cell membranes over time (*Blanchard et al., 2009*; *Butler et al., 2009*; *Lye et al., 2015*), while the *sqh-GFP* signal was used to quantify Myosin II fluorescence intensities for each cell-cell interface identified by cell tracking (*Figure 1A*, *Video 1*). We synchronised movies from 6 embryos to the start of GBE, using our previously described measure of tissue strain rate in the anteroposterior (AP) axis (*Butler et al., 2009*) (*Figure 1—figure supplement 1*). This allowed us to average the Myosin II fluorescence intensities associated with apical cell-cell junctions (interfaces) across embryos, which increased from the start of GBE as expected (*Figure 1B*).

We further extracted independent measures of bidirectional and unidirectional Myosin II planar polarities in the orientation of the AP axis (*Figure 1—figure supplement 2*). Bidirectional polarity of Myosin II (an enrichment at both anterior and posterior cell-cell interfaces for a given cell, *Figure 1C, C'*) was detectable just before the onset of extension and then peaked very early (at 10 min) before declining gradually (*Figure 1C"*), consistent with previous studies (*Kasza et al., 2014*). In contrast, unidirectional polarity (an enrichment in Myosin II at either anterior or posterior cell-cell interfaces, *Figure 1D,D'*) increased progressively for most of GBE (*Figure 1D"*). This suggests that there is a transition from bidirectional to unidirectional Myosin II polarisation over the course of GBE.

To ask whether actomyosin polarities are patterned across the AP axis, we generated spatiotemporal heat maps for both types of polarity for the 6 embryos, as a function of time and position along the AP axis (maps for a representative embryo in *Figure 1E,F*; see other embryos in *Figure 1—*

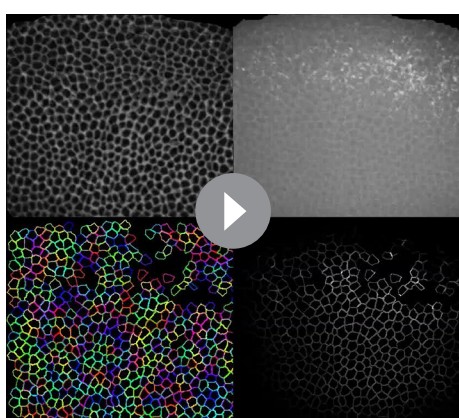

**Video 1.** Representative *sqh^AX3^; sqh-GFP; gap43-Cherry* movie (SG_4), showing the red (top left) and green (top right) fluorescence channels as well as the tracked cell shapes (bottom left) and the quantification of Myosin II fluorescence at tracked interfaces (bottom right). See also **Figure 1A**.

figure supplement 3). Note that in these maps, the data for all cells along the dorsoventral (DV) axis, within an AP bin of defined width, are averaged. Myosin II bidirectional enrichment is strong across the whole AP axis from just before the start of extension until about 20 min, then fades away (green signal in **Figure 1E**). In the unidirectional polarity maps, cells with posterior interfaces enriched in Myosin II (positive values, magenta signal) are distinguished from those where the enrichment is at anterior interfaces (negative values, green signal) (**Figure 1F**). A juxtaposition of opposing unidirectional polarities along the AP axis (magenta next to green signal) thus indicates that shared interfaces between neighbouring cells are enriched in Myosin II. Although the signal is noisy for single embryos, many such juxtapositions are found (**Figure 1F**). These motifs follow the movement of the tissue as it extends towards the posterior (white guide lines in **Figure 1F**). The most prominent ones occur at a regular spacing (arrows in **Figure 1F**). We hypothesised that those correspond to early cable-like enrichments of Myosin II at parasegmental boundaries (PSBs) (**Monier et al., 2010**).

## Parasegment boundaries become mechanically active soon after GBE onset

To test this, we tracked PSBs using two different approaches. First, we identified PSBs from clear cable-like enrichments of Myosin II at the end of the 6 movies analysed above, 60 min after the start of GBE (arrows in **Figure 2A**). Using these boundaries, we manually assigned a parasegment identity to each tracked cell (**Figure 2B**), which could be followed back to the beginning of each movie. This identified PSB cell-cell interfaces at each time point (**Video 2**). We also identified the cell-cell interfaces one-cell diameter anterior and posterior to each PSB (named '-1' and '+1' interfaces, respectively) over time (**Figure 2C,D**). We then quantified the amount of Myosin II found at these three columns of interfaces over time for 3 to 4 parasegments per embryo, for all 6 embryos (**Figure 2E**). We found that the enrichment in Myosin II at PSB interfaces becomes stronger than in the flanking columns of interfaces by 10–15 min of extension. If these cell-cell interfaces enriched in Myosin II were interconnected, they would be expected to straighten, a signature of line tension as shown for other tissue boundaries (**Umetsu et al., 2014**; **Monier et al., 2010**; **Aliee et al., 2012**; **Landsberg et al., 2009**; **Fagotto, 2014**; **Calzolari et al., 2014**). To test this, we quantified the proportion of interfaces oriented between 60 and 90 degrees relative to the AP axis (thus DV-oriented), for each class (**Figure 2—figure supplement 1A,J**). We find that PSB interfaces are more DV-oriented compared to flanking -1 and +1 interfaces, throughout most of GBE (note that all interfaces become briefly very DV-oriented at the beginning of GBE, which is caused by mesoderm invagination transiently stretching the germband cells along DV, see **Lye et al., 2015**). We interpret this as evidence that PSB interfaces align more than flanking interfaces. Together with the preferential enrichment in Myosin II at PSBs (**Figure 2E**), this suggests that PSB interfaces are under higher tension than flanking interfaces during GBE.

To confirm this, we performed laser ablations to probe tension at specific cell-cell interfaces (**Figure 2F–I**) (**Rauzi et al., 2008**; **Farhadifar et al., 2007**). We ablated interfaces located at the PSBs at 40 min (identified by their enrichment in Myosin II, see Materials and methods) and compared them with the ablation of +1 interfaces (one cell diameter posterior to PSB). We checked that PSB and +1 interfaces selected for ablation did not have significantly different lengths (**Figure 2—figure supplement 1G**). PSB interfaces had more Myosin II than +1 interfaces, as expected (**Figure 2—figure supplement 1H**). PSB interfaces are also more DV-oriented than +1 interfaces (**Figure 2—figure supplement 1I**) as expected from our interface alignment analysis. We found that PSB

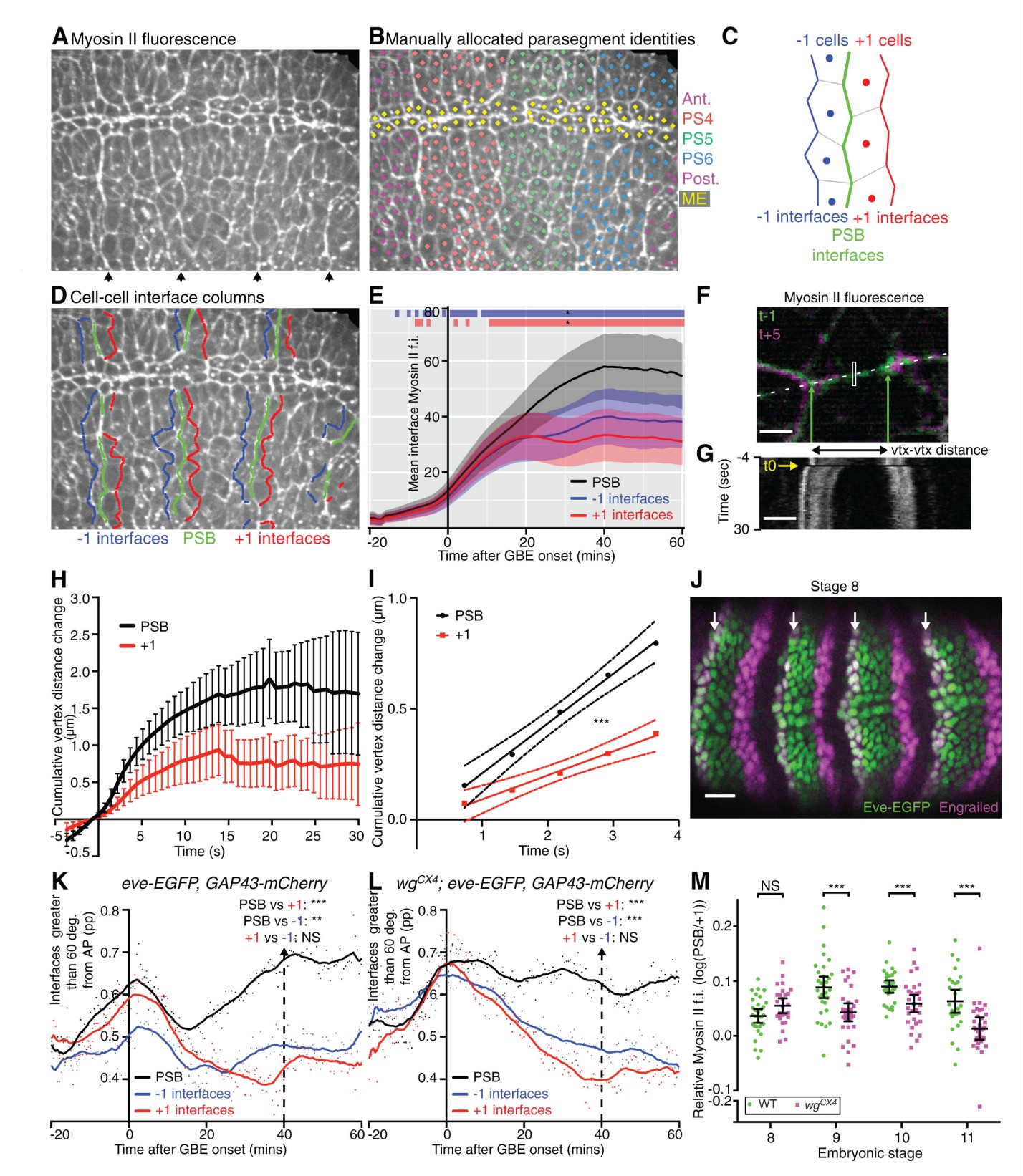

**Figure 2.** Parasegmental boundaries become mechanically active early during axis extension. (A, B, D) Frames of a representative *sqh^AX3^; sqh-GFP; GAP43-mCherry* movie (SG_6) at 60 min after the start of GBE. (A) PSBs are identified at the end of the movie by strong enrichments in actomyosin

*Figure 2 continued on next page*

*Figure 2 continued*

(arrows). (**B**) These are used to manually identify each parasegment (differently coloured cell centroids). Note that the mesectodermal cells (ME, highlighted in yellow) present at the midline are not included in our analyses. (**C,D**) Using parasegment identification, we define 3 classes of linked columns of interfaces, the PSB interfaces (green) and those one cell anterior (named '-1', in blue) and posterior (named '+1', in red) to the PSB, shown in a schematic (**C**) and on the representative movie frame (**D**). (**E**) Myosin II fluorescence intensities (y-axis) found at the three different classes of interfaces over time (x-axis) for the six *sqh^AX3^; sqh-GFP; GAP43-mCherry* embryos. Solid lines represent means. Ribbons (error bands) show an indicative confidence interval of the mean, calculated as a sum of the variance of the embryo means and the mean of the within-embryo variances. Blue and red bars at the top of the panel show time intervals over which -1 and +1 interfaces, respectively, differ from the PSB. Significance is calculated for each one-minute bin using a mixed model ('lmer4' package in 'R') using variation between embryos as the random effect. We use p<0.0005 as the significance threshold, which corresponds to a 0.05 threshold (*) modified by a Bonferroni correction to take into account the 81 one-minutes bins. The same conventions for displaying confidence intervals and statistical significance are used in all subsequent ribbon plots. (**F-I**) Comparison of junctional tension at PSB and +1 cell-cell interfaces using laser ablation. (**F**) Overlay of a PSB junction immediately prior to (-1 time point, green) and after ablation (+5 time point, magenta). The rectangle shows the ablated region. Green arrows show the position of the vertices flanking the junction just prior to ablation. White dashes indicate the line used to produce the kymograph in (**G**). Scale bar, 3µm. The kymograph shows the vertices recoil after ablation (black frame indicated by yellow arrow at time zero). Time corresponds to -3.65 to 29.95 s relative to ablation. The changes in distance between vertices as measured on similar kymographs for each ablation are plotted in (**H**). The graph shows the mean change in vertex distance over time for ablations at PSB (black) and +1 (red) interfaces (N=19 ablated junctions for each). Error bars represent the 95% confidence interval of the mean. (**I**) Graph showing linear regression (solid lines) for the first 5 time points after ablation. The 95% confidence interval of the regressed line is also shown (dotted lines). The data did not significantly deviate from linearity. Slopes were significantly different, with gradients of 0.2245 (+-0.02665) for PSBs and 0.1084 (+-0.0201) for +1s, so a ratio of 2.07 between the two. (**J**) Immunostaining of an *eve-EGFP* embryo at stage 8 using α-GFP and α-Engrailed antibodies, showing that the odd-numbered stripes of Engrailed-expressing cells are faithfully labelled by Eve-EGFP. Scale bar=25 µm. (**K**) -1, PSB and +1 interfaces were identified in the three *eve-EGFP, GAP43-mCherry* movies and their orientation relative to the AP embryonic axis measured. The graph shows the proportion (pp) of interfaces oriented between 60 and 90 degrees relative to the AP axis, as a function of time. A LOWESS curve with a smoothing window of 10 points has been fitted to the data, for this graph and all other interface alignment graphs. Statistical comparisons are shown for the time point 40 min (Cumulative interface orientation distribution for all interfaces at 40 min are shown in *Figure 2—figure supplement 1K*). The convention for P values for this graph and all subsequent similar graphs are: NS: p>0.05; *p<0.05; **p<0.01; ***p<0.001. (**L**) Same analysis for 3 *wg^CX4^; eve-EGFP, GAP43-mCherry* movies (See also *Figure 2–supplement 1L*). This shows that in *wingless* mutants, PSB interfaces are more DV-oriented than -1 or +1, as in wildtype (**K**). (**M**) Graph comparing Myosin II enrichment at PSBs relative to +1 interfaces in fixed embryos labelled with α-Sqh1P antibodies, during GBE (stage 8) and at later stages (stage 9, 10, 11), in wildtype and *wingless* mutants. Input data and statistics are in *Figure 2—source data 1*.

The following source data and figure supplement are available for figure 2:

**Source data 1.** Source data for *Figure 2*, including statistical analysis.

**Source data 2.** Source data for *Figure 2—figure supplement 1*, including statistical analysis.

**Figure supplement 1.** Identification and characterisation of parasegmental boundaries properties during axis extension.

vertices recoiled significantly faster than +1 interfaces and to a greater extent (*Figure 2H*). We estimated the difference in recoil velocities to be a factor of 2 (*Figure 2I*). This confirms that PSB interfaces are under higher tension than flanking interfaces and validates our interface alignment analysis.

To further confirm that PSBs are mechanically active during axis extension, we used a second approach to identify these boundaries, using *eve-EGFP* (*Venken et al., 2009*) to directly label the PSBs in embryos expressing *GAP43-mCherry* (the latter to track cell interface behaviours as before). We found that *eve-EGFP* reliably marks the anterior edge of odd-numbered parasegments throughout GBE (*Figure 2J*, *Figure 2—figure supplement 1B–F*). This allowed us as before to assign parasegment identities to cells and to track the PSB and flanking -1 and +1 interfaces at

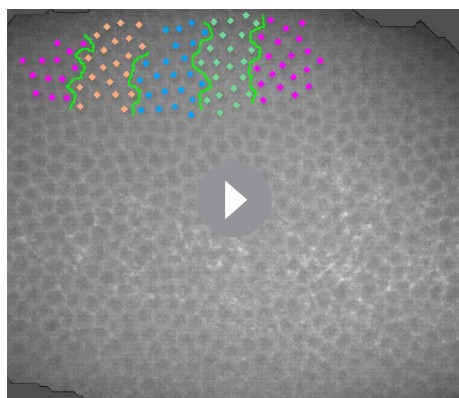

**Video 2.** Representative *sqh^AX3^; sqh-GFP; gap43-Cherry* movie (SG_4) showing the green channel (*sqh-GFP*) with identification of the different parasegments and the parasegmental boundary interfaces. See also *Figure 2A,B,D*.

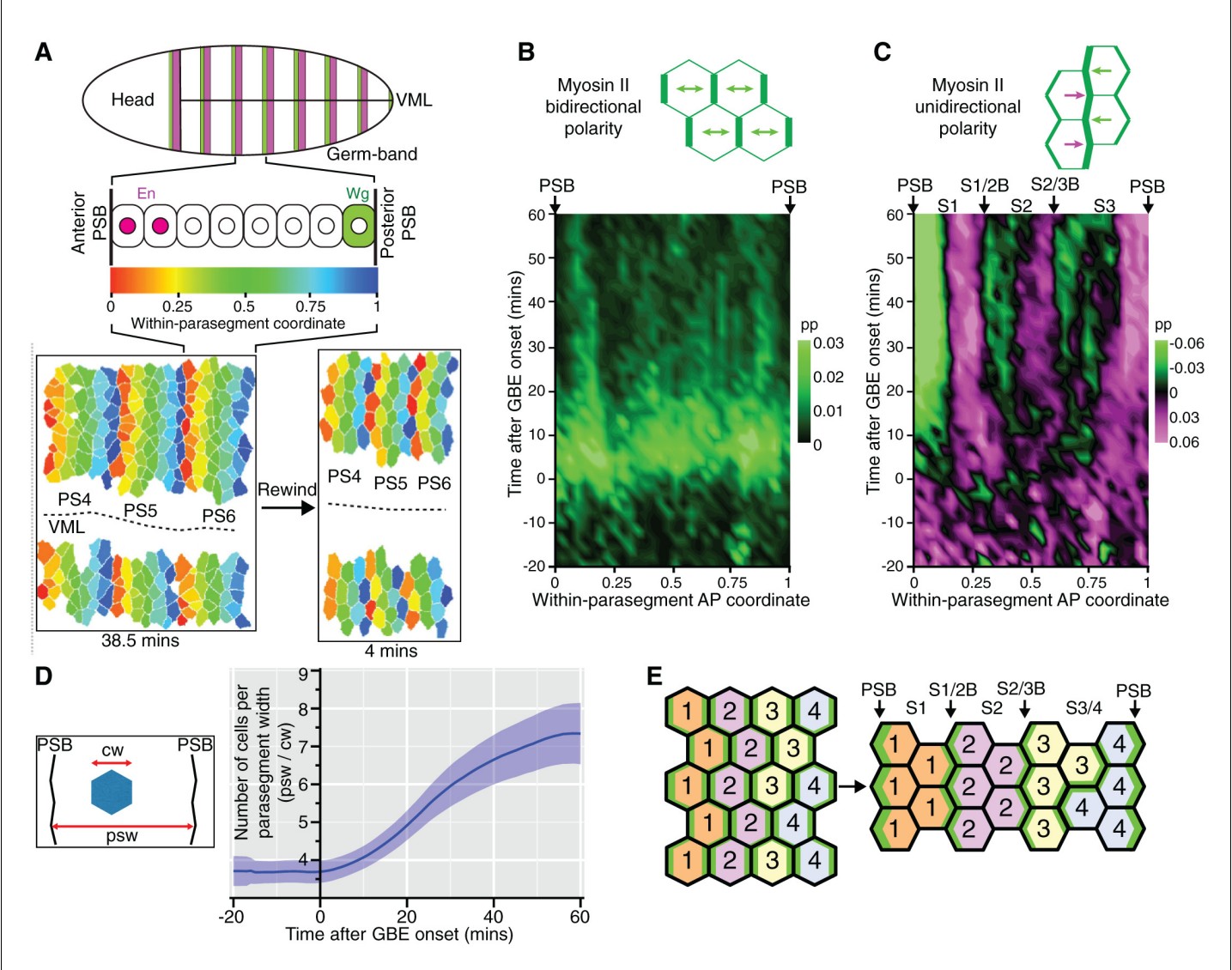

**Figure 3.** Within-parasegmental patterns reveal two further myosin-enriched boundaries at stereotypical AP locations. (**A**) Schematic of *Drosophila* embryo showing the parasegment domains along AP (VML: ventral midline). Cells expressing Engrailed (En) and Wingless (Wg) abut the posterior and anterior edge, respectively, of each parasegmental boundary (PSB). The identification of PSBs in movies (*Figure 2*) was used to allocate an AP coordinate to each cell within each parasegment domain. The anterior-most position is recorded as 0 (red in the heat scale) and the posterior-most position is recorded as 1 (blue). This coordinate system is used to pool cell information from all the different parasegments present throughout each movie, in order to look for stereotypical within-parasegment patterns. AP coordinates for a representative movie are shown at a late (38.5 min) and an early (4 min) timepoint, for 3 parasegments tracked (PS4, PS5 and PS6). (**B**) Spatio-temporal map showing Myosin II bidirectional polarity for all averaged PS domains, as a function of the within-parasegmental AP coordinate (x-axis) and time relative to the start of GBE (y-axis, in min). Heat scale shows highest bidirectional polarity in bright green and no polarity in black (See statistics in *Figure 3—figure supplement 1A,B*). (**C**) Spatio-temporal map showing Myosin II unidirectional polarity for all averaged PS domains, as a function of the within-parasegmental AP coordinate (x-axis) and time relative to the start of GBE (y-axis, in min). Heat scale shows enrichment of posterior cell-cell interfaces as magenta (positive values) and of anterior ones as green (negative values) (See statistics in *Figure 3—figure supplement 1A,C*). (**D**) Quantification of average cell number per parasegment domain as a function of time relative to the start of GBE (y-axis, in min). Cell numbers are obtained by dividing the average parasegment width (psw) by the average cell width (cw). (**E**) Diagram showing the proposed model: at the start of GBE, 3 to 4 cells of distinct identity per parasegment enrich Myosin II at their shared interfaces. After cell rearrangement, stripes of cells of the same identity become adjacent. Myosin II is enriched preferentially at interfaces shared between cells of different identity (PSBs, S1/2Bs and S2/3Bs, also marked on panel (**C**). There is more Myosin II enrichment at PSBs compared to other boundaries, indicated as thicker green lines. We postulate that the third stripe, S3 as defined by unidirectional polarity data above (panel **C**), is composed of a mixture of two identities, named 3 and 4 here, whose boundary is more variable. In support of this, S3 is wider than S1 and S2, but not wide enough for 4 cells across (2+2) (see also cell numbers per stripe in *Figure 4D*). Input data and statistics are in *Figure 3—source data 1*.

*Figure 3 continued on next page*

*Figure 3 continued*

The following source data and figure supplement are available for figure 3:

**Source data 1.** Source data for *Figure 3*, including statistical analysis.
**Source data 2.** Source data for *Figure 3—figure supplement 1*, including statistical analysis.
**Figure supplement 1.** Within-parasegmental patterns of Myosin II cell polarity.

odd-numbered PSBs through time, for 3 *eve-EGFP, GAP43-mCherry* movies. We confirmed that interface orientation differences between PSB and flanking interfaces were replicated in these movies, where PSBs are labelled without relying on their enrichment in Myosin II (*Figure 2K* and *Figure 2—figure supplement 1K*). Together, these results show that PSB interfaces are mechanically active by 15–20 min at the latest after GBE onset, much earlier than their previously known role at stage 10 when they segregate dividing boundary cells (*Monier et al., 2010*). Since cell division in the germ-band ectoderm does not commence until 40 min after GBE onset in our movies (*Figure 1—figure supplement 1B*), this suggested that PSBs have an early mechanical role during polarised cell intercalation.

Because later in development, Myosin II enrichment at PSBs depends upon Wingless (Wnt-1 homologue, expressed in one row of cells immediately anterior to the PSB interfaces; *Monier et al., 2010*; *Sanson, 2001*), we asked if this signalling pathway was also required for the mechanical activity of the PSBs during GBE. To test this, we generated 3 movies expressing *eve-GFP* and *GAP43-mCherry* in a *wingless* null mutant background ($wg^{CX4}$; *eve-EGFP, GAP43-mCherry* embryos). We performed the same interface orientation analysis as before, and found that the PSBs straightened in *wingless* mutant embryos as in wildtype (compare *Figure 2K and L* and *Figure 2—figure supplement 1K and L*). We also quantified Myosin II enrichment at PSBs (relative to +1 interfaces) in fixed embryos at stages 8 to 11 (*Figure 2M* and *Figure 2—figure supplement 1M–M'''*). Although Myosin II is significantly decreased in *wingless* mutants at PSBs once the germband has finished extending (stages 9, 10 and 11), confirming our previous findings (*Monier et al., 2010*; *2011*), we found no difference during GBE (stage 8). We conclude that the selective enrichment in Myosin II at PSB interfaces and their straightening during GBE is not controlled by Wingless, suggesting that it is under pair-rule gene control.

## Unidirectional polarity patterns are a consequence of polarized cell intercalation

There were more unidirectional polarity patterns in our spatiotemporal maps than just those corresponding to PSBs (*Figure 1F* and *Figure 1—figure supplement 3*). To characterise those, we increased the resolution of our maps by averaging the data collected for each of the 6 $sqh^{AX3}$; *sqh-GFP; GAP43-mCherry* movies. We used our identification of PSB interfaces to attribute a within-parasegment coordinate value to each cell from 0 (anterior-most) to 1 (posterior-most) over time (*Figure 3A* and *Video 3*). Using this coordinate system, we averaged data from 3 to 4 parasegments per movie for our 6 movies. We replotted bidirectional and unidirectional polarity patterns at this parasegmental scale (*Figure 3B, C*). Confirming individual movie maps (*Figure 1E* and *Figure 1—figure supplement 3*), we found that AP-oriented bidirectional Myosin II polarisation is strong across parasegmental domains until

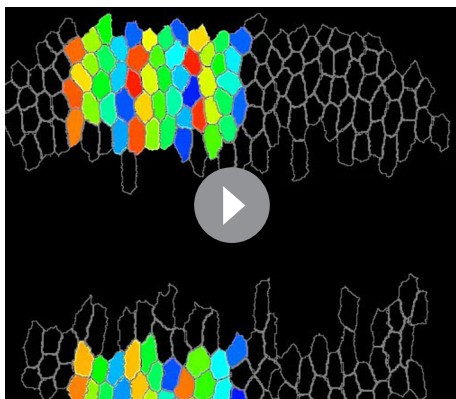

**Video 3.** Representative $sqh^{AX3}$; *sqh-GFP; gap43-Cherry* movie (SG_4) showing the tracked cell contours with within-parasegment coordinate colour-coded as shown in *Figure 3A*.

about 15 min after extension, decreasing thereafter (bright to dark green signal in *Figure 3B*; statistics in *Figure 3—figure supplement 1A,B,D*). In contrast, unidirectional polarity emerges gradually from the start of GBE (*Figure 3C*; *Figure 1F*; statistics in *Figure 3—figure supplement 1A,C,E*). First, as expected, anterior and posterior interfaces at PSBs have strong Myosin II enrichments of opposite sign, from as early as 10 min after GBE onset (green and magenta respectively at each edge of the plot in *Figure 3C*). Second, the increase in resolution reveals two more positions along the AP axis where anterior and posterior unidirectional polarities alternate (magenta/green boundaries highlighted with arrows in *Figure 3C*; statistics in *Figure 3—figure supplement 1C*). This suggests that there are columns of interfaces in at least two stereotypical locations within each parasegment that become enriched in Myosin II. This gradual transition from global bidirectional polarities to precisely located unidirectional polarities suggested that new DV-oriented junctions not enriched in Myosin II form as a consequence of cell rearrangements. To monitor the progress of cell rearrangements, we quantified the number of cells across the parasegmental domains over time. The average number of cells per parasegment width (along AP) almost exactly doubles, from 3.6 cells at the start of axis extension to 7.3 after 60 min (*Figure 3D* and *Figure 3—figure supplement 1F*). This shows that the emergence of unidirectional polarity is concurrent with the progress of polarised cell intercalation.

To explain these patterns, we propose the following model. Because of the precision of the segmentation cascade (*Dubuis et al., 2013*; *Tkačik et al., 2015*), it is conceivable that there are as many cell identities as there are cells per parasegment width (3–4 on average, see *Figure 3D* and model in *Figure 3E*). At the start of GBE, actomyosin enrichment would occur at each cell-cell interface based on these differences in identity along the AP axis. When cells intercalate and make new contacts, this would bring cells of the same identity adjacent to each other along AP. Because their identities are the same, their new shared interfaces would not enrich in Myosin II (*Figure 3E*). In contrast, interfaces between stripes of cells of different identity would continue to enrich in Myosin II, driving the emergence of persistent unidirectional polarity. A corollary of this model is that Myosin II polarisation is a consequence of local cell-cell interactions rather than global signals. If a global mechanism was at play, actomyosin would be expected to be enriched at all new DV-oriented interfaces, maintaining bidirectional polarisation, which is not what we find (*Figure 3B,C*).

## S1/2 and S2/3 boundaries in each parasegment enrich Myosin II and straighten during axis extension

This model generates specific predictions that we can test. In particular, the two new columns of interfaces identified as having strong unidirectional polarity within each parasegment should have more Myosin II and straighten more than the intervening cell-cell interfaces, after they emerge through cell intercalation. We tracked these, by manually identifying junctions enriched in Myosin II at the end of each movie (as previously done for the PSBs), at the AP locations mapped in our spatiotemporal plots (*Figure 3C*). This initial analysis defined 3 stripes per parasegment (S1, S2 and S3, *Figure 4A* and *Video 4*) and identified cell-cell interfaces separating stripes 1 and 2 (S1/2B) and stripes 2 and 3 (S2/3B) (boundary interfaces), from cell-cell interfaces within each stripe (non-boundary interfaces) (*Figure 4B*, see also *Figure 3C,E*). We checked that the S1/2B and S2/3B interfaces identified at the end of GBE had AP positions consistent with their expected boundary positions throughout the movies (*Figure 4C*). Next, we checked that cell numbers for each stripe matched those expected from the model, with S1 and S2 increasing approximately from 1 to 2 cells wide, and S3 from 1.5 to 3 cells wide, from start to end of GBE (*Figure 4D*). The larger width of S3 is explained in our model: S3 would be composed of a mixture of cell identities 3 and 4, because there are not enough cells in a parasegment (3.6 cells on average rather than 4 at the onset of GBE, *Figure 3D*) to make a two-cell stripe for either identities 3 or 4 at the end of GBE (*Figure 3E*).

Next, we quantified Myosin II enrichment at the within-parasegment boundaries. As predicted, interfaces belonging to the boundaries S1/2B and S2/3B become more enriched in Myosin II than interfaces immediately anterior (-1) or posterior (+1) (*Figure 4E,F*). We then examined the orientation of the different classes of interfaces over time. As predicted, S1/2B and S2/3B are more DV-oriented than +1 or -1 control interfaces (*Figure 4G–J*). Note that as expected from the unipolarity maps (*Figure 3C*), S1/2B and S2/3B are less enriched in Myosin II and less DV-oriented than the PSBs (see PSB curves shown for comparison in *Figure 4E–H*), but overall these three boundaries have comparable behaviours. Based on our analysis of cell number, Myosin II enrichment and

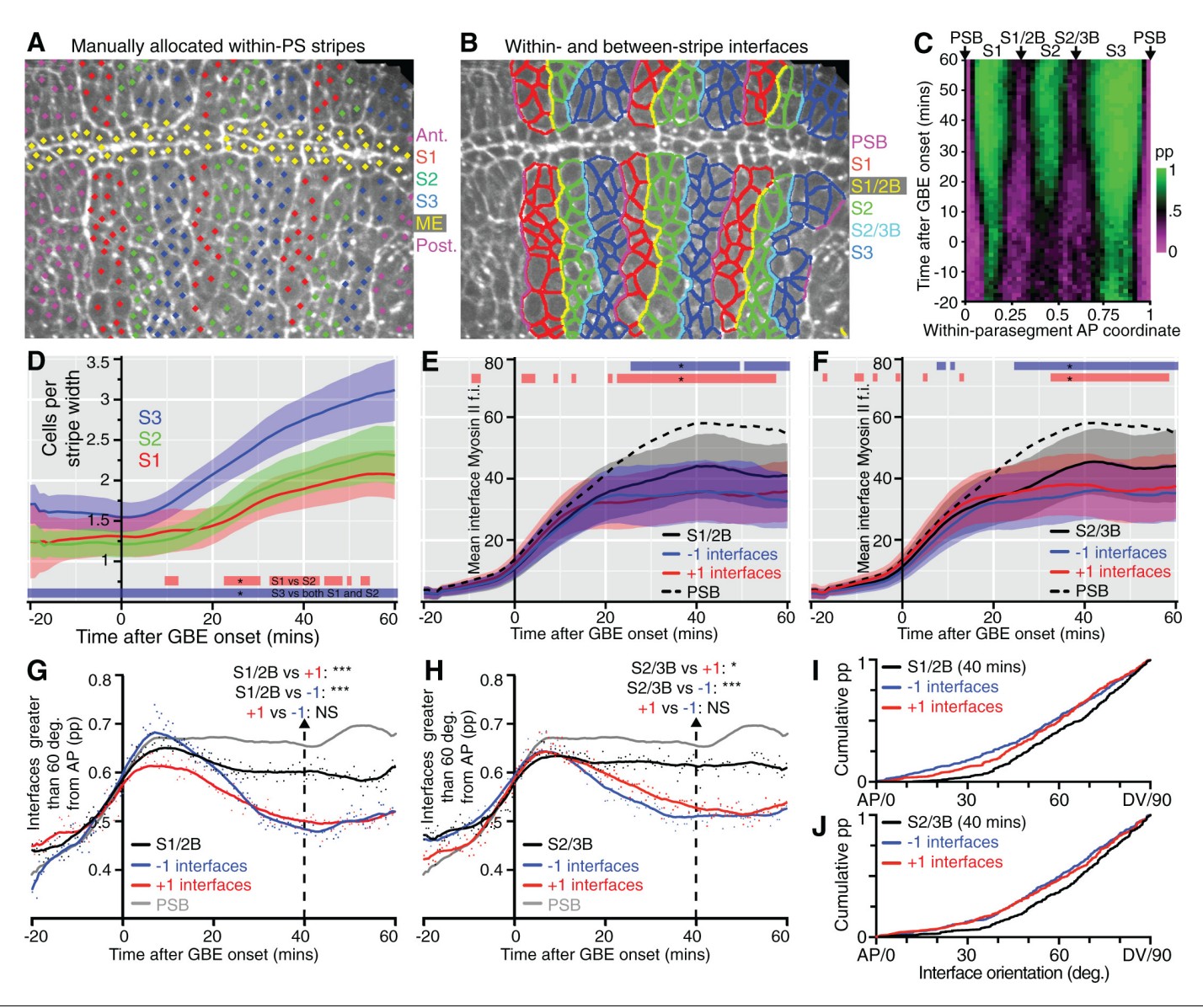

**Figure 4.** Behaviour of S1/2 and S2/3 boundaries. (**A**) The image is taken from a *sqh^AX3*; *sqh-GFP*; *GAP43-mCherry* movie 60 min after the start of GBE, where cells have been manually allocated to putative within-parasegment stripes S1 (red centroids), S2 (green centroids) and S3 (blue centroids), based on Myosin II enrichment and position along AP. In this movie, allocation was done for 3 parasegments (magenta centroids highlight cells belonging to other parasegments and yellow centroids belong to midline cells, ME). (**B**) Same movie frame where interfaces are classified as belonging to boundaries between stripes (PSB interfaces in magenta, S1/2B interfaces in yellow, S2/3B interfaces in cyan) or not belonging to any boundaries (red interfaces in S1, green interfaces in S2 and blue interfaces in S3). (**C**) Spatiotemporal plot (time in y-axis and within-parasegment coordinates in x-axis) to check that the locations of manually identified within-parasegment boundaries correspond to the location of the S1/2B and S2/3B given by the unidirectional polarity map (arrows, see *Figure 3C*). The proportion (pp) of non-boundary interfaces is colour-coded so that 1 is green (only non-boundary interfaces) and 0 is magenta (only boundary interfaces). There is high concordance between the locations of S1/2B and S2/3B interfaces in both plots (compare with *Figure 3C*). (**D-J**) Once stripe and interface identities are allocated, analyses can be performed on all tracked parasegments throughout GBE. (**D**) Average cell number per stripe in AP (y-axis) as a function of time from the start of GBE (x-axis). At the bottom of the panel, red bar indicates the time intervals where S1 differs from S2, and blue bar where S1 and S2 differ from S3. (**E**) Average Myosin II intensity at boundary interfaces between stripe 1 and 2 (S1/2B) compared to interfaces immediately anterior (-1) or immediately posterior (+1). Mean for PSB interfaces is shown for reference (dashed line). Blue and red bars at the top of the panel show time intervals where -1 and +1 interfaces, respectively, differ from S1/2B interfaces. (**F**) Same quantifications as in E but for S2/3B. (**G**) Proportion of interfaces with orientation between 60 and 90 degrees relative to the AP axis (y-axis), as a function of time (x-axis), for S1/2B interfaces compared to -1 or +1 interfaces. The same measure for PSB interfaces is shown for

*Figure 4 continued on next page*

*Figure 4 continued*

reference (grey curve). A statistical comparison is shown at 40 min (see also I). (**H**) Same quantifications as in **G**, but for S2/3B. (**I, J**) show the cumulative distributions of interface orientation for S1/2B and S2/3B and control interfaces at 40 min. Input data and statistics are in *Figure 4—source data 1*.

The following source data is available for figure 4:

**Source data 1.** Source data for *Figure 4*, including statistical analysis.

interface orientation, we conclude that we have identified two new columns of interfaces enriched in Myosin II within parasegments, with the behaviour predicted by our model (*Figure 3E*).

## Boundary and non-boundary interfaces have distinct behaviours during axis extension

A further prediction of our model is that Myosin II enrichment should respond to the juxtaposition of different cell identities rather than to the orientation of the cell-cell interfaces relative to the main embryonic axes. To test this prediction, we examined Myosin II enrichment at boundary interfaces (PSBs, S1/2B and S2/3B) relative to non-boundary interfaces, as a function of interface orientation relative to the AP axis. Before 25 min of GBE, boundary interfaces have more Myosin II than non-boundary interfaces for all orientations except those parallel to AP (0 to about 20 degrees) (left panel in *Figure 5A* and *Figure 5—figure supplement 1A–C'*). For both types of interface, there is some dependency upon orientation, with higher enrichment for interfaces closer to 90 degrees relative to AP (DV-oriented interfaces), consistent with previous studies (see for example Figure 4D in *Kasza et al., 2014*). This dependency upon orientation is lost after 25 min, with boundary interfaces strongly enriched compared to non-boundary interfaces, irrespective of orientation (right panel in *Figure 5A* and *Figure 5—figure supplement 1A–C'*). We conclude that although some more global mechanism might contribute to Myosin II enrichment at the beginning of GBE, cell-cell interactions dominate overall.

Another prediction from our model is that the boundary interfaces should drive convergence extension, in other words they should shorten actively, since they are more enriched in Myosin II than non-boundary interfaces. We have already shown that PSBs, S1/2Bs and S2/3Bs become straighter than intervening interfaces, which is evidence that they are more contractile. To ask if they participate more in cell rearrangements, we developed a method to capture the cell neighbour exchanges called T1 transitions (see Materials and methods). T1 transitions are identified by following the shrinkage of a given interface and linking it to the growth of a new interface (*Figure 5B*). Using this method, we identified every T1 transition occurring in stripes S1 and S2 for all tracked parasegments in our 6 *sqh^AX3^; sqh-GFP; GAP43-mCherry* movies. We did not analyse stripe S3 as we cannot unambiguously identify boundary interfaces separating the putative cell identities 3 and 4 in that stripe. For each T1 transition identified, we have information on how much Myosin II is found at shortening and elongating interfaces. Pooling all the T1 transitions in S1 and S2 together, we find that Myosin II increases with interface shortening prior to the interface swap (*Figure 5C*), consistent with prior studies (see for example *Figure 1g,i* in *Rauzi et al., 2010*). To distinguish between the interfaces that are shortening actively from those that may shorten passively, we developed another method to probe geometric stress (see Materials and methods). We assume that a Voronoi tessellation based on cell centroid locations represents a mechanically neutral configuration for the cell-cell interfaces. We measured the deviation in interface length from this

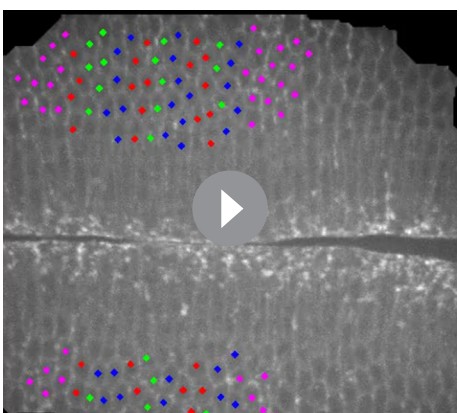

**Video 4.** Representative *sqh^AX3^; sqh-GFP; gap43-Cherry* movie (SG_4) showing the within-parasegment stripes colour-coded as in *Figure 4A*.

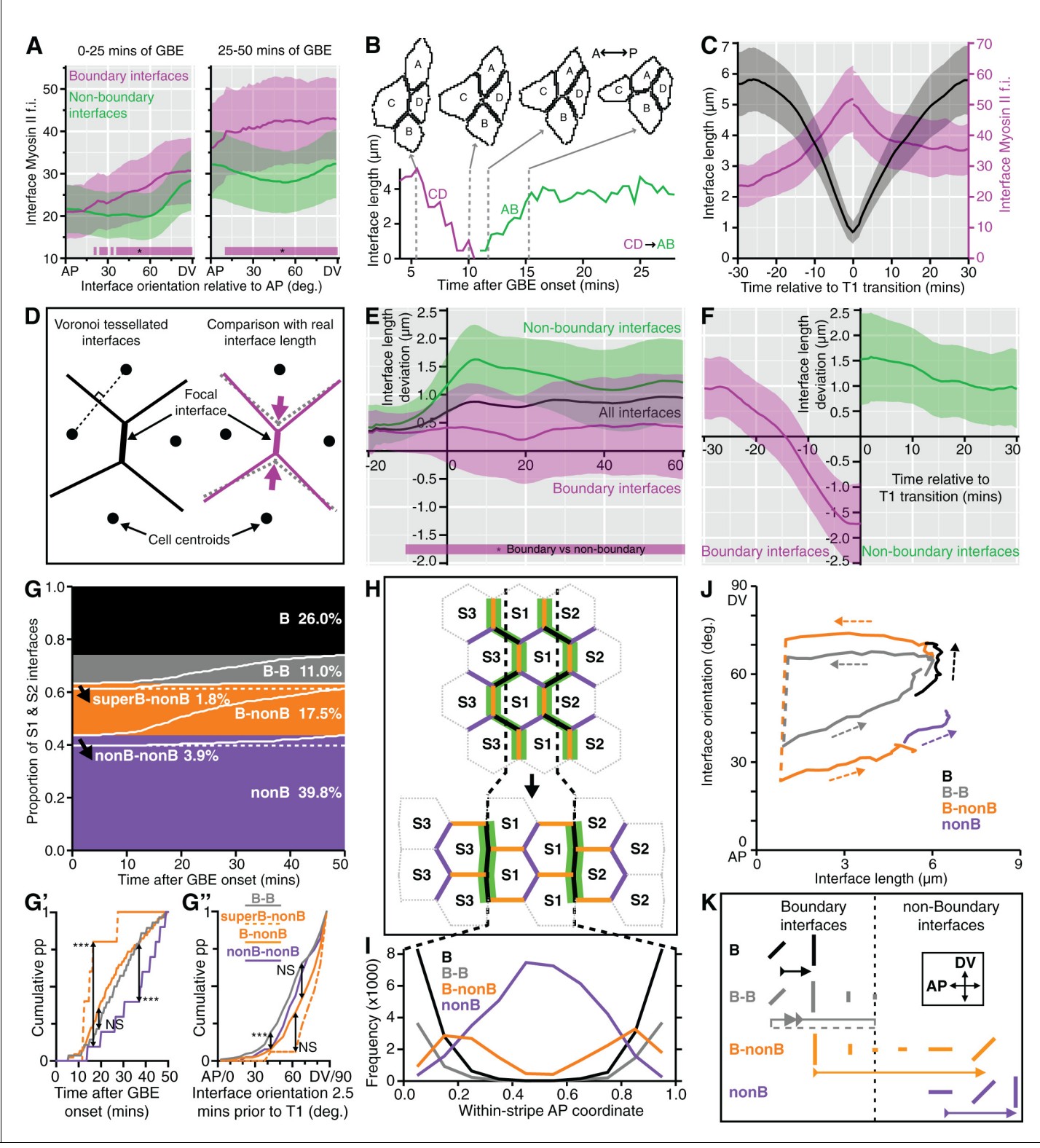

**Figure 5.** Characterisation of the behaviours of boundary and non-boundary interfaces. (**A**) Average Myosin II intensity in boundary versus non-boundary interfaces for two time periods of GBE (0–25 and 25–50 min), as a function of their orientation relative to the AP embryonic axis. 0 degrees is parallel to AP, 90 degrees parallel to DV. (**B**, **C**) Analysis of cell neighbour exchanges. (**B**) Example of a T1 transition where the interface between cells C and D shortens to a single vertex, followed by the growth of a new interface between cells A and B. The graph gives the interface length (y-axis) as a function of time after the start of GBE (x-axis). In this particular example, the T1 transition starts at 5 min and finishes at 15 min after the start of GBE. *Figure 5 continued on next page*

*Figure 5 continued*

(C) Aligning all interfaces in time so that the T1 transitions are at zero min, this plot shows how the shortening of interfaces (black curves) correlates with the increase in Myosin II fluorescence intensity (magenta curves) during neighbour exchange. (D-F) Analysis of cell geometries. (D) We compared interface lengths predicted by a Voronoi tessellation (black on the left, dotted grey on the right) with real interface lengths (magenta) to extract a length deviation from the Voronoi tessellation, a geometric proxy for local stress. (E) Graph showing the average deviation in length from a Voronoi prediction (y-axis), for all interfaces (black line), for boundary interfaces (magenta curve) and for non-boundary interfaces (green curve), as a function of GBE time (x-axis). Non-boundary interfaces are on average longer and boundary interfaces shorter than the average length deviation for all interfaces. (F) On average, boundary interfaces become increasingly geometrically stressed (shorter than Voronoi prediction) over a period of 15 min just prior to T1 transitions. (G-K) Fate of boundary (abbreviated to B) and non-boundary (abbreviated to nonB) interfaces during GBE, for stripes S1 and S2 (Data pooled from 6 embryos, N=96,343 interface instances). (G) S1 and S2 interfaces behaviours fall into four main types: interfaces that remain boundary throughout GBE and do not go through a T1 transition (black); interfaces that remain boundary throughout but go through a T1 transition (grey); boundary interfaces that go through a T1 transition and become non-boundary (orange); interfaces that remain non-boundary interfaces throughout (purple). The percentage of each interface type is shown. Within each type, interfaces are sorted according to the time of T1 transition (white lines). Black arrows indicate two infrequent subtypes. In the orange class, a subtype of boundary interfaces corresponds to interfaces between either cell identities 1 and 3 (cell identity 2 is missing) or 3 and 2 (cell identity 1 is missing). We call these interfaces 'super-boundaries' (abbreviated to superB) (see main text). We have inferred that identity 1 or 2 are skipped because for this subtype the tracking data shows that either stripe S1 or stripe S2 has a local width of zero. The other subtype is in the purple class (arrow) and corresponds to rare non-boundary interfaces that do go through a T1 transition. (G') Comparison of the timings of T1 transitions in the different interface types. The two infrequent subtypes have opposite behaviours: the super-boundary T1 transitions (dashed orange curve) are earliest, while non-boundary T1 transitions are latest (purple curve) compared to boundary T1 transitions (orange and grey). (G'') Comparison of the distributions of the orientations of interfaces 2.5 min prior to T1 transition for the different types of interface. 'Super-boundary' interfaces are the most DV-oriented (dashed orange). (H) Cartoon showing the expected location of the four types of interfaces relative to the position of the stripe boundaries (dashed black lines). Green shows Myosin II enrichment. (I) Graph giving the frequency (y-axis) of each type of interface as a function of the AP position within a S1 or S2 stripe (x-axis). Each AP location (bin) within a stripe is attributed a within-stripe coordinate from 0 (anterior-most) to 1 (posterior-most). (J) Plot showing the fates of each type of interface during GBE. The mean interface orientation (y-axis) and length (x-axis) is plotted for each type over time. Dashed arrows show the direction of time. Dashed lines connect interfaces before and after T1 transitions. See *Figure 5—figure supplement 1E* for individual curves for each *sqh^AX3; sqh-GFP; GAP43-mCherry* movie. (K) Cartoon summarising the behaviour of each type of interface during GBE. Changes in length and orientation of interfaces are depicted as well as the transition between boundary and non-boundary class. Direction of time is indicated by arrows. The dashed part of the grey arrow depicts the situation where a boundary interface remains a boundary after a T1 swap. Input data and statistics are in *Figure 5—source data 1*.

The following source data and figure supplement are available for figure 5:

**Source data 1.** Source data for *Figure 5*, including statistical analysis.
**Source data 2.** Source data for *Figure 5—figure supplement 1*, including statistical analysis.
**Figure supplement 1.** Analysis of cell-cell interface behaviour.

tessellation for boundary and non-boundary interfaces (*Figure 5D*, *Figure 5—figure supplement 1D* and Materials and methods). We find that boundary interfaces are shorter than predicted by a Voronoi tessellation (*Figure 5E*), particularly so in the 15 min prior to a T1 transition (*Figure 5F*), indicating that they actively shorten during GBE. We conclude that the boundary interfaces that we have identified drive convergence of the germ-band.

Next, we examined the behaviour of all interfaces during GBE for S1 and S2 (*Figure 5G–K*). We identify four main interface behaviours. About a quarter of interfaces are boundary interfaces which are not involved in any T1 transitions and remain boundary interfaces throughout GBE (black in *Figure 5G–K*). At the start of GBE, these interfaces are oriented on average about 50 degrees relative to the AP axis, then rotate to become oriented closer to DV, around 70 degrees (*Figure 5J,K* and *Figure 5—figure supplement 1E*). Another quarter of interfaces are boundary interfaces involved in T1 transitions, with two distinct behaviours: some remain boundary interfaces after the T1 swap, while others become non-boundary interfaces (grey and orange, respectively, in *Figure 5G–K* and *Figure 5—figure supplement 1E*). Finally, the rest of the interfaces are non-boundary interfaces which, for their large majority, are not involved in T1 transitions as expected (purple, *Figure 5G–K* and *Figure 5—figure supplement 1E*). This confirms that boundary interfaces are those involved in cell neighbour exchange. Each interface behaviour occurs at the expected AP locations within each stripe, giving further support to our model (*Figure 5H,I*).

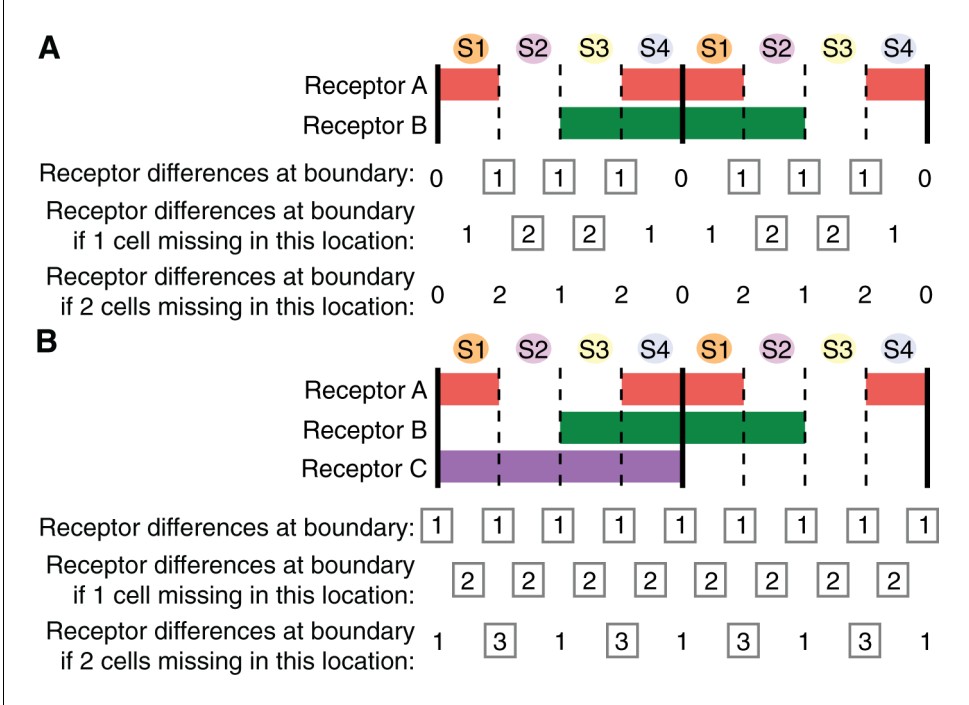

**Figure 6.** Finding the smallest number of receptors explaining Myosin II planar polarization during axis extension. (**A**) Expression patterns of two putative receptors **A** and **B** repeated every double parasegment (corresponding to the expression patterns of, respectively, Toll-2 and Toll-8 as described in Figure 1p in *Paré et al., 2014*). PSBs are shown as solid black lines. Summing the number of receptor differences at each boundary, this combination lacks a difference at the PSBs. For cell pairs brought together when single cells are missing (second line), the number of cell receptor differences increases only when cell identity 2 or cell identity 3 is missing (grey boxes highlight an increase in receptor differences). There is no increase in receptor differences, hence no robustness, built in if two contiguous cells are missing at any location (third line). We calculate a 'robustness' score by adding the number of instances, for two parasegments, where there is an increase in receptor differences in the event of 1 or 2 cells missing: the score for this scenario is 10 (number of grey boxes for a double parasegment unit) (see *Figure 6—figure supplement 1C*). (**B**) When considering 0, 1 or 2 missing cells, the most robust solution with three receptors is achieved with an additional receptor C spanning one parasegment out of two (either odd or even). This provides a receptor difference at the PSBs (grey boxes in first line) and systematically increases receptor differences when one cell is missing at a given location (grey boxes in second line). When 2 cells are missing, the number of receptor differences increases at a subset of locations, notably in the case where cell identities 3 and 4 are missing (grey boxes in third line). The robustness score for this solution is 20 (arrow in *Figure 6—figure supplement 1C*). Code for receptor permutations is in *Source code 1*.

The following source data and figure supplement are available for figure 6:

**Source data 1.** Source data for *Figure 6—figure supplement 1*, including statistical analysis.
**Figure supplement 1.** Combinatorial receptor patterns.

---

Furthermore, examining now the whole data set (considering all three stripes S1, S2 and S3), we find that the number of T1 transitions consistently peaks at the expected locations for PSBs, S1/2B and S2/3B (*Figure 5—figure supplement 1F*). Interestingly, we also find a T1 transition peak in the middle of S3 (blue curves in *Figure 5—figure supplement 1F,F'*), which would correspond in our model to an incomplete or variably located boundary between cell identities 3 and 4 (*Figure 3E*). This is corroborated by a peak in Myosin II in the middle of stripe S3 (*Figure 5—figure supplement 1G*). Using an independent measure of cell intercalation (intercalation strain rate, see (*Butler et al., 2009*; *Blanchard et al., 2009*) and Materials and methods), we find that the rate of intercalation is higher in stripe S3, compared to stripe S1 and S2 (*Figure 5—figure supplement 1H*). We think that this higher rate of intercalation in stripe S3 is caused by missing cells of identity 3 or 4 in this stripe.

We postulate that when a cell identity is missing in the AP parasegmental sequence, such as cell identity 3 or 4, the resulting interface enriches more Myosin II and consequently intercalates faster and earlier that other interfaces. We expect these 'superboundary' interfaces (behaving as 'superintercalators') to be most prevalent in stripe S3 because of insufficient cells there, but our data suggest that these can be found also (but rarely) in stripe S1 and S2 (SuperB subtype in *Figure 5G–G"*). We conclude that the variable number of cells per parasegment along AP causes a faster intercalation rate in the posterior part of the parasegment compared to the anterior part (*Figure 5—figure supplement 1H*).

## Modelling the minimum number of receptors required for the planar polarisation of Myosin II during axis extension

The current molecular explanation for the planar polarization of Myosin II during GBE is that pair-rule genes control the expression in stripes of three Toll-like receptors that provide a heterotypic code for the enrichment of Myosin II at AP cell-cell interfaces (*Paré et al., 2014*). The code is thought to be incomplete because it currently does not explain interface enrichment at PSBs (*Paré et al., 2014*). Here we asked what is the minimum number of receptors that could explain all of the Myosin II patterns that we have uncovered in this study. We first considered a scenario recapitulating as closely as possible the expression of the three Toll-like receptors (Toll-2, Toll-6 and Toll-8) identified in *Paré et al. (2014)*. We noted that Toll-6 and Toll-8 were largely interchangeable (*Paré et al., 2014*). Therefore our first scenario has a receptor A and a receptor B respectively expressed in pair-rule patterns broadly similar to Toll-2 and Toll-6/8 (*Figure 6A*). Assuming initially 4 cells per parasegment, we counted by how many receptors adjacent cells differed along the AP axis. For example, if a cell expresses a receptor and the adjacent cell does not, then we recorded a difference of 1 for the corresponding AP interface (*Figure 6A*). We postulate that a difference of one receptor or more triggers Myosin II enrichment at the corresponding interface. In this first scenario, all interfaces along AP differ by one receptor, except at the PSBs where there are no differences, consistent with the conclusion that the Toll-like receptor patterns currently do not explain Myosin II enrichment at PSBs (*Paré et al., 2014*).

We then considered what happens when one cell is missing in the sequence of four cell identities along AP in each parasegment. We know this has to be frequently the case since we find an average of 3.6 cells per parasegment at the start of GBE (*Figure 3D*). We counted again the number of receptors at interfaces, when a cell is missing at a given position. For example, if cell identity 2 is missing, cell identities 1 and 3 become adjacent; since cell identity 1 is expressing receptor A and cell identity 3, receptor B, we scored a difference of two receptors for this particular interface (*Figure 6A*). Remarkably, we find that the number of receptor differences increases by one in many locations when a cell identity is missing (*Figure 6A*). We predict that the number of receptor differences is likely to be proportional to the amount of Myosin II recruited. In other words, we propose that the receptor identity system is quantitative. If the amount of Myosin II enriched is indeed proportional to the number of receptor differences, then more rapid cell intercalation would be expected to occur where cells are missing in the AP sequence (see 'superboundaries' and 'superintercalators' introduced earlier). Increased cell intercalation would fill the gaps in cell identity during GBE and maintain the cell order along AP. From our data, the cell identities that are most likely to be missing are 3 and 4, since we find that there are not enough cells to make two columns of two cells at the end of GBE in stripe 3 (*Figure 3D,E and 4D*). For example, according to the scenario in *Figure 6A*, if cell identity 3 is missing, the receptor numbers at adjacent interfaces 2/4 increases from 1 to 2 (*Figure 6—figure supplement 1A*). This in turn should translate into an increase in Myosin II at those interfaces, which then would increase the rate of intercalation. This notion is supported by our data, since we find that the cell intercalation rate is higher in stripe 3 containing identities 3 and 4, than in stripes 1 or 2 (*Figure 5—figure supplement 1H*).

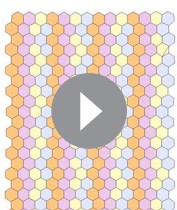

**Video 5.** Movie of simulation 4 shown in *Figure 7G*.

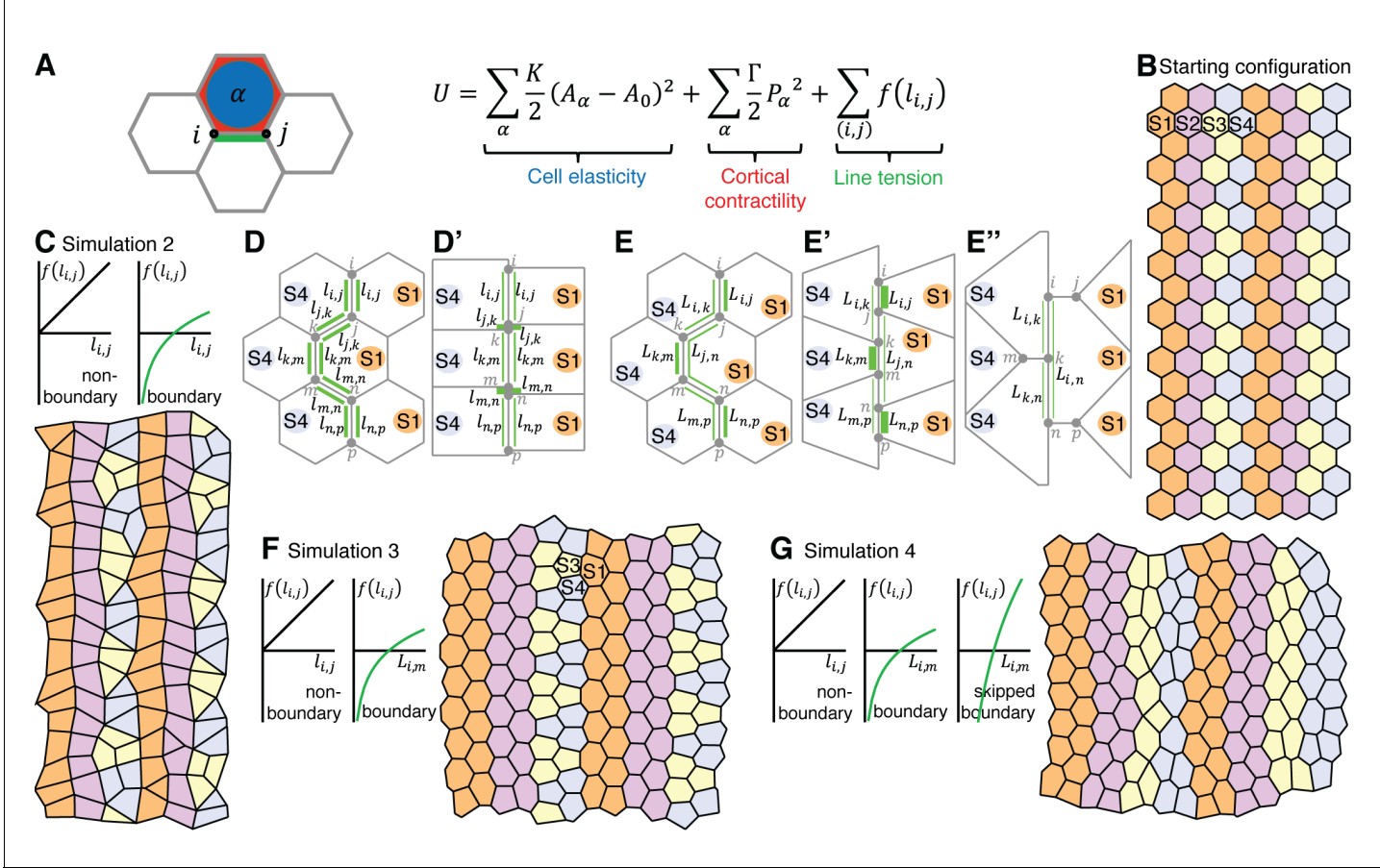

**Figure 7.** A vertex model based on cell-cell interactions replicates the interface behaviours during axis extension. (A) Summary of vertex model of germband extension. Cells are considered as two-dimensional polygons representing cell-cell interfaces, with vertices forming where three polygons meet. An 'energy function' is calculated and used to update the position of every vertex $i$ over time. This energy function encodes mechanical contributions associated with cell elasticity, cortical contractility and interfacial 'line tension energy'. We consider a number of simulations (see main text), which differ in the hypothesised dependence of the line tension $f$ on interface lengths. (B) The initial configuration for each simulation comprises regular hexagonal cells organised into parasegments, each comprising cells of 4 stripe identities (S1-S4). Note that the initial configuration starts with 14x20 cells (*Video 5*) and smaller snapshots are shown in this Figure and *Figure 7—figure supplement 1*. (C) In simulation 2, the line tension energy $f$ varies linearly with interface length for non-boundary interfaces, but we specify a nonlinear dependence for boundary interfaces to represent a positive feedback between interface shortening and Myosin II enrichment. In this simulation, cells undergo neighbour exchanges but become stuck locally in four-cell junctions and hence convergent extension cannot proceed (D, D'). In Simulation 3, we apply our non-linear dependence of line tension to the total length of contiguous *boundary* interfaces for a given cell (length $L_{i,m}$) (E), rather than to individual boundary interfaces (length $l_{i,j}$) (D, D'). This allows vertices to slide independently on either side of a column of interfaces that makes a boundary (E', E'') and the simulated tissue now undergoes convergent extension (F), but identities S3 and S4 clump together. (G) Simulation 4 resolves this issue by incorporating 'supercontractility', where boundary interfaces between cells of non-adjacent identities ('skipped boundary') are more contractile. Code for vertex model is in *Source code 1*.

The following source data and figure supplement are available for figure 7:

**Source data 1.** Source data for *Figure 7—figure supplement 1*, including statistical analysis.

**Figure supplement 1.** Further details of vertex model simulations for interface behaviours during axis extension.

From the above, we propose that the receptor system is robust to missing cells because gaps in the pattern will be 'repaired' by speeding up intercalation at cell-cell interfaces most different in their receptor composition. Building on this hypothesis, we looked for the most likely expression pattern for a third putative receptor that would both explain the enrichment at the PSBs but also confer enhanced robustness to missing cells. To do this, we explored all possible permutations of three receptors, where each is expressed in a putative pair-rule pattern (four cell-stripes out of eight, in a

given double parasegment unit) (*Figure 6B*). We scored each permutation by summing both the number of immediate neighbour receptor differences, and also the *increase* in receptor differences at each interface if one or two cell identities are missing in a given row of cells. The permutation that scored highest (20, see *Figure 6—figure supplement 1C*) expresses two receptors in the same exact pattern as our first scenario (*Figure 6A*) and a third receptor in every other parasegment (*Figure 6B*). With this solution, the number of receptors at cell-cell interfaces increases systematically from one to two when a cell identity is missing anywhere in a double parasegment unit, therefore showing robustness. When two cells are missing, this number increases to three receptors at a subset of locations. Interestingly, one of these locations corresponds to the case where cell identities 3 and 4 are both missing (*Figure 6B*; *Figure 6—figure supplement 1B*). Since our data suggest that identities 3 and/or 4 are those most likely to be absent (1 and 2 being more systematically specified), this solution confers adequate robustness, taking into account the observed polarity of the parasegment. Note that a solution with 4 receptors instead of 3 does show a better robustness throughout the double parasegment unit (*Figure 6—figure supplement 1D,E*), but since cell identities 1 and 2 are less likely to be missing according to our data, we conclude that the three-receptor solution shown in *Figure 6B* is the most parsimonious.

## A vertex model based on cell-cell interactions replicates the interface behaviours during axis extension

To test our cell identity model more formally, we implemented a vertex model with a starting configuration of 20 rows and 14 columns of regular hexagonal cells, organised into 4 parasegments, with each parasegment comprising 3 to 4 cell identities along AP (*Video 5*). In vertex models, the movement of junctional vertices is governed by the strength of cell-cell adhesion, the contractility of the actomyosin cortex and cell elasticity (*Farhadifar et al., 2007*; *Fletcher et al., 2014*; *Honda and Eguchi, 1980*). These contributions are encoded in a 'free energy' function, whose gradient determines the velocity of each vertex. In addition, cell neighbour exchanges (T1 transitions) occur whenever a cell-cell interface's length falls below a threshold value. We use a free energy function based on (*Farhadifar et al., 2007*) (*Figure 7A*), keeping the 'cell elasticity' and 'cortical contractility' terms the same throughout, but varying the 'line tension energy' term in successive simulations to model different features of interface contractility inferred from the real data (*Figure 7—figure supplement 1A*).

In simulation 1, the line tension energy associated with each cell-cell interface varies linearly with its length, but the boundary interfaces are twice as contractile as the non-boundary interfaces (*Figure 7—figure supplement 1B*). Cells fail to undergo neighbour exchange in this simulation. In simulation 2, the linearity of line tension energy is replaced at boundary interfaces by a non-linear relationship, where the line tension energy decreases at an ever-faster rate as the interface shortens. This models a positive feedback between interface shortening and Myosin II enrichment, supported by our data (*Figure 5C*). In this simulation 2, cells do now undergo neighbour exchanges, but become stuck in a four-cell junction topology (*Figure 7C–D'*). In simulation 3, we allow vertices to slide independently on either side of a column of interfaces that makes a boundary (*Figure 7E–F*). We implement this (for boundary interfaces only) by applying our non-linear dependence of line tension to *all* interfaces present at a given boundary for a given cell (combined length $L$) (*Figure 7E–F*), rather than to individual interfaces (length $l$) (*Figure 7C–D'*). The cells are now able to intercalate and the simulated tissue undergoes convergent extension, elongating in AP while shortening in DV (*Figure 7F*). As a consequence, single columns of cell identities 1 and 2 become double columns of cells at the end of the simulation, as predicted in *Figure 3E*. However, because of their insufficient number, cells of identities 3 and 4 end up clumping together according to their identity, thereby disrupting the AP order of the starting pattern (*Figure 7F*).

To address this, we implemented a fourth simulation (*Figure 7G*) that incorporates our hypothesised 'supercontractility', where interactions between cells of non-adjacent identities in the parasegmental sequence generate more contractile interfaces than cells of adjacent identities. For example, contractility would be higher at interfaces between identities 2 and 4, than between 2 and 3 or 3 and 4. Implementing this, simulation 4 solves the clumping problem and maintains the AP order of cell identities throughout axis extension (*Figure 7G*), as postulated in *Figure 3E*. So simulation 4 recapitulates the intercalary cell behaviours that we hypothesise based on our data. Finally, we analysed interface behaviours as for the real data (*Figure 5I*). We find that boundary and non-boundary

interfaces in simulation 4 have behaviours qualitatively similar to real data (Compare *Figure 7—figure supplement 1C* with *Figure 5I*), demonstrating that this simulation successfully models the cell interface behaviours of GBE.

## Discussion

We have developed new computational methods to quantify and analyse patterns of Myosin II planar polarisation and cell behaviours in the extending *Drosophila* germband in both time and space. In previous studies, the analysis of Myosin II planar polarity has focused on bipolarity, often comparing the enrichment in Myosin II at the DV-oriented sides (also called vertical sides) of germband cells relative to their AP-oriented sides (also called horizontal sides) (for example, see *Simões et al., 2014*; *Paré et al., 2014*; *Kasza et al., 2014*). Here, in addition to using a measure of bidirectional polarity, we have developed a measure of unidirectional polarity, to identify when one side of a cell is enriched relative to all other sides. By distinguishing between bi- and unidirectional polarities, we have been able to identify novel patterns that inform how Myosin II planar polarisation arises and drives cell and tissue behaviours. Furthermore, by taking a live-imaging approach, we have been able to observe how these polarities evolve with unprecedented temporal resolution.

Our study provides further experimental evidence that differential cell identity generates the planar polarity of Myosin II in the germband and extends existing models. A long-standing hypothesis in *Drosophila* segmentation is that the cascade of genes from maternal determinants, gap genes and then pair-rule genes is able to establish differential 'identities' with single-cell precision along the AP axis (*Dubuis et al., 2013*; *Tkačik et al., 2015*). The discovery of a role for Toll-like receptors, under the control of pair-rule genes in GBE, has provided compelling molecular evidence for this model (*Paré et al., 2014*). One question arising from this work is what happens to Myosin II planar bipolarity once polarized cell intercalation proceeds. Indeed, polarized cell intercalation will increase the cell number along AP, thus bringing cells with the same identity next to each other along this axis. If differential cell identity via heterotypic interactions drives Myosin II polarisation throughout GBE, then some cells should find themselves in homotypic interaction with either an anterior or posterior neighbour, which would not lead to Myosin II enrichment (See *Figure 3E*). The unipolarity patterns that we find are consistent with this hypothesis, identifying alternating domains of enriched and not enriched cell-cell interfaces along AP, which emerge during the course of axis extension. These correspond to Myosin II-enriched boundaries between parasegmental domains (PSBs) and to at least two more locations within each parasegment from early in GBE. The AP position of these enrichments is consistent with these being the consequence of the doubling of cell numbers along AP via polarized cell intercalation. Therefore the differential identity model predicts a transition between bidirectional and unidirectional polarities over the course of GBE, which is validated by our data.

Another prediction of the differential identity model is that Myosin II enrichment should be dependent upon the type of cell-cell interface (homotypic versus heterotypic) rather than interface orientation (DV versus AP-oriented). We were able to test this by comparing the orientation of enriched boundary interfaces (heterotypic in our model) versus non-boundary interfaces (homotypic). Early GBE (0–25 min) is characterised by two features. First, as predicted by a cell-cell interaction model, boundary interfaces are significantly more enriched than non-boundary interfaces for most orientations. However, overlaid on this, DV-oriented interfaces are also more enriched in Myosin II, irrespective of their boundary/non-boundary identity. This relationship between interface orientation and Myosin II enrichment in early GBE is at odds with a model based solely on cell-cell interactions. It is unclear what the cause of this relationship might be. Some planar polarity and cell intercalary behaviours remain in mutants for all three Toll-like receptors identified (*Paré et al., 2014*). A possibility is that the remaining polarity is due to a more distant polarising signal operating in early embryos, which would direct Myosin II to all DV-oriented interfaces. Later in GBE (25–50 min), our analysis shows that Myosin II enrichment becomes independent of interface orientation, indicating that distant polarising signals are not acting on the germband at this stage and that local cell-cell interactions dominate.

The Toll receptor model proposed in *Paré et al. (2014)* relies on each parasegment being four cells wide. Our quantification shows that parasegments are in fact on average only 3.6 cells wide at the onset of GBE (sampling parasegments 4 to 7, see Materials and methods). The widths of the

stripes containing cell identities 1 and 2 are consistent with single-cell wide columns increasing to two-cell wide columns and therefore behave as expected from the differential cell identity model. However, the distinction between the stripes containing cell identities 3 and 4 as predicted by *Paré et al. (2014)* was less clear. Instead we observe a third stripe, which is 1.5 cells wide in AP on average at the start of GBE, increasing to 3 after 60 min. We think it likely that the cell types 3 and 4 do exist as postulated by *Paré et al. (2014)*, since there are detectable peaks of Myosin II and neighbour exchanges in the middle of our third stripe (*Figure 5—figure supplement 1F,G*). But because parasegments are less than 4 cells across at GBE onset, some rows would have only cell types 1,2,3 or 1,2,4, while others have the full complement of cell types 1,2,3,4 (see *Figure 3E*). After 60 min of GBE, stripes 3 and 4 would then give a mixture of arrangements, such as 3,3,4 and 3,4,4. As a result, the expected enrichment of Myosin II at heterotypic interfaces between cells of identity 3 and 4 would not align well, explaining why we cannot resolve a stripe 3/4 boundary in our data. If our reasoning is correct, this implies that there is an inherent polarity within each parasegment, with the anterior half made of cell types 1 and 2 being robustly specified, while in the posterior half, specification of cell identities 3 and 4 is more variable. This polarity might be important for the tissue to cope with the variation of cell number across parasegments and to repair the AP patterns during cell intercalation. Indeed, at the start of axis extension, although parasegments are usually 3 or 4 cells across, they occasionally have rows that are fewer or more cells across (*Lawrence and Johnston, 1989*; *Busturia and Lawrence, 1994*). We conclude that the mechanism of active convergence of the germ-band must be robust to variable cell number within each parasegmental unit.

Our modeling suggests a mechanism by which the embryo copes with this variable cell number during axis extension. We postulate that the cell-cell interaction mechanism that triggers Myosin II enrichment at interfaces along AP is quantitative. It has been proposed that the stripy expression of Toll-2, 6 and 8 receptors generate heterotypic interactions that result in Myosin II enrichment (*Paré et al., 2014*). We further propose that these receptors, in addition to at least another receptor at the PSB, produce Myosin II enrichment which is proportional to the strength of the heterotypy. In other words, the more adjacent cells differ in the number of receptors they express, the more Myosin II will accumulate at their shared interfaces. We find that three receptors expressed in a pair-rule pattern is sufficient in theory to explain the planar polarization of Myosin II at every interface along AP in the germband, including the PSB interfaces which were not accounted for by the Toll-2,6,8 combinatorial code (*Paré et al., 2014*). Two of the receptor patterns we identify correspond to the patterns of Toll-2 and Toll-6/8 (Toll-6 has a pattern similar to Toll-8) and the third provides heterotypy at the PSB. The remarkable finding with this minimal combination of receptors is that heterotypy increases when one cell is missing in any position along AP. Moreover, heterotypy increases further when two cells are missing at half of the positions along AP. This is true in particular when identities 3 and 4 are both missing, which are the identities we think are most likely to be absent, based on our data. So when cells are missing, heterotypy would increase, triggering more Myosin II enrichment. This would increase the intercalation rate at the most mismatched interfaces and lead to pattern repair. In support of this, we do find an increased rate of cell intercalation in the posterior part of the parasegment (*Figure 5—figure supplement 1H*), where we predict more mismatches because of too few cells of identities 3 and 4.

We tested these hypotheses in a vertex model and recapitulated qualitatively the tissue-scale behaviours in the data. We had to implement specific interface behaviours in the model to have successful convergence-extension of the *in silico* tissue. These are based on plausible behaviours in vivo. In particular, one limitation of vertex models is that apposed cortices are modeled as a single interface. The changes between Simulation 2 and 3 attempt to go round this limitation: what we tried to model is a situation where cells behave independently on either side of a boundary. For example, junctions could slide independently of each other on either side of the boundary. This is possible in vivo because a boundary is made of two cell cortices, and each cell cortex at the boundary interface could elongate or shorten independently. This could conceivably happen if the two cell cortices on either side of a boundary have different contractile forces. In addition to junctional sliding, cell-cell sliding could occur along the boundary, for example if adhesion is decreased there. Further work is required to determine if these processes are happening during GBE. Another point of note, we have implemented ratios of 1:2:8 for the line tension energies between non-boundary, boundary and 'supercontractile' boundary interfaces in Simulation 4. The 1:2 ratio is quantitatively

consistent with observed ratios of tension between PSB boundary and non-boundary interfaces obtained by laser ablation (*Figure 2I*). We do not know what to expect as a ratio between boundary and supercontractile boundary interfaces, but 8 seems high. A discrepancy between the ratios of tension needed for a successful simulation of boundary behaviour and the ratios estimated in vivo by laser ablation has been noted by *Landsberg et al. (2009)* and so the relationship between line tension energies in vertex models and tension measured by laser ablation might not be simple/linear. For this paper, the key point is that the model qualitatively supports the idea that some boundary interfaces are more contractile than others.

In combination with the two other receptor patterns which would correspond to those of Toll-2 (receptor A in our model) and Toll-6/8 (receptor B), our parsimonious three-receptor combination is in theory sufficient to explain all of the Myosin II polarity patterns we identify in our study. By identifying PSB interfaces at late stages by their strong myosin enrichment and backtracking to earlier in development, we have further demonstrated that the PSB dominates over the two intra-parasegmental boundaries in terms of myosin enrichment. The predominance of the PSB is detectable from very close to the start of GBE. At the onset of GBE, PSBs are already demarcated genetically by the expression of the pair-rule genes such as *eve* and *ftz* and the gradually increasing expression of segment polarity genes *wg* and *en* (*Jaynes and Fujioka, 2004*). However, this is the first time that a cellular (rather than genetic) characteristic has been identified for PSBs this early. After the end of germband extension, later in development when epidermal cells are actively dividing, the movement of dividing cells across PSBs is prevented because the boundary interfaces enrich in Myosin II relative to non-boundary interfaces (*Monier et al., 2010*), as for other compartmental boundaries in *Drosophila* (*Umetsu et al., 2014*; *Aliee et al., 2012*; *Landsberg et al., 2009*; *Major and Irvine, 2006*) and for tissue boundaries in zebrafish (*Calzolari et al., 2014*) and *Xenopus* (*Fagotto, 2014*; *Fagotto et al., 2013*). In all these cases, the enrichment in Myosin II has been proposed to increase interfacial tension and promote tissue segregation. A possibility is that the PSBs fulfill a similar role during GBE, to prevent mixing between adjacent parasegments that cell intercalation might cause otherwise. Our interface orientation analyses and our laser ablation experiments demonstrate that there is indeed an increase in interfacial tension at PSBs early in GBE. We propose that elevated line tension at PSBs and also, to a lesser extent, at the two intra-parasegmental boundaries that we have identified, contribute to maintain the AP sequence of cell identities while cell rearrangements are occurring.

It is unclear why the PSB boundaries are enriching Myosin II more than the other two intraparasegmental boundaries we have identified. This could be explained if the heterotypy between cell identity 1 on the posterior side of the PSB is strongest in combination with cell identities 3 or 4 on the anterior side of the PSB. We predict that a not yet identified receptor, with a pattern of expression corresponding to receptor C in our most parsimonious model (*Figure 6B*), directs myosin II recruitment at the PSB interfaces. It could be that this putative receptor triggers a stronger response at the PSBs compared to the Toll-like 2,6,8 receptors at the other boundaries. Alternatively, more than one receptor might be contributing heterotypy at the PSBs. Our data suggest that we can rule out an early role for Wingless signaling in contributing to a PSB-specific response. Indeed, while Wingless is required to maintain Myosin II enrichment at the PSB later in development (*Monier et al., 2010*; *2011*), it is not required for the enrichment *during* germ-band extension (*Figure 2M*), which is corroborated by the fact that PSBs straighten in *wingless* mutants as in wild-type (*Figure 2L*). Thus it is likely that the pathway directing strong enrichment of Myosin II specifically at PSBs is under pair-rule control.

Finally, our analysis shows that cell interface behaviour associated with active intercalation predominantly occurs at the boundary interfaces that we identify. Thus in *Drosophila* GBE, intraparasegmental boundaries and PSBs enriched in actomyosin appear to drive GBE. Supracellular Myosin II cables are already known to drive tissue elongation through the formation of multicellular rosettes (*Blankenship et al., 2006*). These have not been linked to specific positions along the AP axis, but it is likely that rosettes form exclusively at PSBs or intraparasegmental boundaries, where our analysis suggests that Myosin II is enriched in continuous cable-like structures.

In conclusion, we think that we have identified segmentally repeated boundaries, which enrich in Myosin II and simultaneously drive cell intercalation while keeping cells ordered along the AP axis. Our findings contribute to the growing evidence that cell fate heterogeneities are translated into differential interface contractility to govern morphogenesis (*Paré et al., 2014*; *Bielmeier et al., 2016*;

*Bosveld et al., 2016*). Extending the work of *Paré et al. (2014)*, we propose an updated differential cell identity model that is robust to missing cells, postulating a third receptor expressed in every other parasegment as the most parsimonious solution. In *Xenopus,* the antero-posterior patterning of the mesoderm also drives convergent extension (*Ninomiya et al., 2004*), and thus similar ordering mechanisms might operate in vertebrate systems. As a whole, this system is reminiscent of the 'self' versus 'non-self' recognition mechanisms thought to play a role during neuronal wiring in the nervous system (*Zipursky and Grueber, 2013*; *He et al., 2014*), and might represent a more ancient and primitive 'non-self' avoidance system, co-opted here by AP patterning to control cell behaviours. The logic and rules that we have uncovered for *Drosophila* axis extension provides a paradigm for more complex structures such as the brain (*Hassan and Hiesinger, 2015*).

## Materials and methods

### *Drosophila* strains

We used the null mutants $sqh^{AX3}$ (*Jordan and Karess, 1997*) and $wg^{CX4}$ (*Baker, 1987*) and the transgenes *en-lacZ* (on II) (*Busturia and Morata, 1988*), $sqh\text{-}GFP^{42}$ (on II) (*Royou et al., 2004*), *GAP43-mCherry* (on III) (*Martin et al., 2010*) and *eve-EGFP* (on III) (*Venken et al., 2009*) to construct the stocks $sqh^{AX3}$; $sqh\text{-}GFP^{42}$; *GAP43-mCherry/TM6B*, *yw;;eve-EGFP,GAP43-mCherry/TM6B* and *w; $wg^{CX4}$, en-lacZ/CTG. $yw^{67}$*embryos were used as WT. The CTG balancer chromosome was *CyO, twi-GAL4, UAS-GFP* (*Halfon et al., 2002*).

### Immunohistochemistry and imaging of fixed *Drosophila* embryos

We followed standard methods for fixing and staining *Drosophila* embryos, using the primary antibodies goat anti-GFP (ab6662, Abcam, 1:200), rabbit anti-Engrailed (d300, Santa Cruz Biotechnology; 1:50), rabbit anti-β-gal (ECK0341, MP Biomedicals; 1:2500), rat anti-DE-CAD (DCAD2, DSHB; 1:50), guinea pig anti-Sqh-1P (*Zhang and Ward, 2011*; 1:100). We used the following secondary antibodies: goat anti-rabbit-Alexa-594, goat anti-rat-Alexa-594 and goat anti-guinea pig-Alexa-488 (Life Technologies, 1:500). To improve immunostaining against Sqh-1P, embryos were post-fixed in 4% formaldehyde for 15 min before secondary antibody staining.

Embryos were mounted individually on slides in VECTASHIELD (Vector Labs) under a coverslip suspended by a one-layer thick magic tape (Scotch) bridge on either side. This flattened the embryos sufficiently so that all cells were roughly in the same z-plane. Prior to placing the coverslip, embryos were rolled so that their ventral surfaces were facing upwards towards the coverslip. Embryos were imaged on a Nikon Eclipse TE2000 inverted microscope incorporating a C1 Plus confocal system (Nikon). Images were captured using Nikon EZ-C1 software. Optical z-stacks were acquired with a depth of 0.25 µm between successive optical z-slices and with a total optical z-stack depth sufficient to capture both the top of the embryo and any more basal markers of the parasegment boundary (PSB). All embryos were imaged using a violet corrected 60x oil objective lens (NA of 1.4). Laser illumination at 488 nm wavelength was used for Alexa-488 fluorophores and 543 nm for Alexa-594. Neutral density (ND) 4 filters were applied to all lasers. Recursive averaging of 4 was used. The gain and offset were optimized for each embryo.

### Semi-automated quantification of cortical Myosin II in fixed embryos

For each stage and each genotype quantified ($yw^{67}$ or *w; $wg^{CX4}$, en-lacZ*), 9–10 embryos (1–4 boundaries per embryo) were analysed. Embryos were immunostained for Sqh-1P (as a marker of Myo II), DE-CAD (as a marker of cell membranes) and a PSB marker (En or βgal, depending on the embryo genotype). Quantification was performed on PSB interfaces and +1 interfaces, which could all be identified relative to the position of PSB marker staining.

Connected interfaces, in which Myo II was to be quantified, were traced using the FIJI plugin Simple Neurite Tracer (*Longair et al., 2011*) based on DE-CAD staining. Where possible, a line of interconnected interfaces was traced between the ventral midline and the amnioserosa. If a region of dividing cells was encountered along one of these lines of interconnected interfaces, the tracing was stopped and restarted the other side of the dividing cells. The traced lines were then increased in width by one pixel each side, giving a total line width of three pixels. Quantification was performed in the Sqh-1P channel. Fluorescence values lower than the modal pixel intensity were subtracted as

background fluorescence. Average fluorescence intensity was calculated for each 3-pixel wide line trace using ImageJ. PSB interface fluorescence intensity was then normalised to +1 interface fluorescence intensity on a per PSB basis. Statistics were performed in Prism (GraphPad).

## Live imaging of *Drosophila* embryos

Embryos were dechorionated in commercial bleach before being rinsed thoroughly in water. An oxygen permeable membrane was pulled tightly over a custom-made metal imaging insert. Nine stage 5 embryos were mounted, ventral-side towards the objective, on the membrane in halocarbon oil (Voltalef PCTFE, Atofina, France) in a 5 mm spaced 3x3 array. A coverslip was placed over the embryos, supported by a bridge of a single coverslip on each side.

Embryos were imaged under a 40x oil objective lens (NA of 1.3) on a Nikon Eclipse E1000 microscope with a Yokogawa CSU10 spinning disk head and a Hamamatsu EM-CCD camera. Embryos were illuminated using a Spectral Applied Research LMM2 laser module (491 nm and 561 nm excitation). Images were captured using Volocity Acquisition Software (PerkinElmer). Embryos were positioned under the objective lens so that the field of view was slightly posterior to the point at which embryos were widest in their DV axis. Optical z-stacks of a thickness of 28 μm were captured, with 14 μm above the top of the embryo and 14 μm into the embryos at the beginning of acquisition (to allow for movement of the embryo in the z-axis). Consecutive optical z-slices were separated by 1 μm. Embryos were imaged every 30 s from late stage 5 for 100 min. Movies were recorded at 20.5 ± 1°C, measured with a high-resolution thermometer (Checktemp1).

To check that embryos survived the imaging process to the end of embryogenesis. *sqh^AX3^*; *sqh-GFP*; *GAP43-mCherry* and *eve-EGFP, GAP43-mCherry* embryos were allowed to develop on the imaging insert to hatching in a humidified box. *wg^CX4^*; *eve-EGFP, GAP43-mCherry* embryos were treated similarly, but because *wingless* mutants are embryonic lethal, the cuticle of embryos was prepared using standard methods to check their phenotype. Occasional movies acquired for embryos that did not hatch or did not make a cuticle at the end of embryogenesis were discarded.

## Cell tracking

The confocal z-stacks were converted into stacks of curved quasi-two-dimensional representations, the outermost of which followed the surface of the embryo with deeper layers shrinking progressively in 0.5 μm steps towards the centre of the embryo. The section giving the clearest view of cell apices was selected for tracking. Bespoke tracking software identifies cells and links them in an iterative process using an adaptive watershedding algorithm (*Blanchard et al., 2009*; *Butler et al., 2009*). For each cell at each time point, coordinates of cell centroids, perimeter shapes, cell-cell interfaces, and links forwards and backwards in time for both cells and interfaces (even through neighbour exchange) are stored.

No statistical method was used to predetermine embryo number. We previously tracked cells in 5 embryos per treatment in *Butler et al. (2009)*, which was sufficient to show treatment differences. WT morphogenesis is remarkably reproducible (see *Figure 1—figure supplement 1A,B*) so we considered 6 *sqh^AX3^*; *sqhGFP*; *GAP43-mCherry* embryos would be sufficient to show robust patterns. For *eve-EGFP, GAP43-mCherry* and *wg^CX4^*; *eve-EGFP, GAP43-mCherry* embryos, we performed manual correction of segmented cell outlines at all time points. This improved the tracking in the embryos, hence we required only 3 embryos per treatment (summarized in *Table 1*).

## Movie synchronisation in space

Movie x and y pixel coordinate axes were rotated and transformed into embryonic AP and DV coordinates in μm. The origin of embryonic coordinates was set at the start of GBE as the intersection

**Table 1.** Summary of embryos analysed per genotype.

| Embryo Genotype | # Movies analysed | Mode of Tracking |
| --- | --- | --- |
| *sqh^AX3^*; *sqh-GFP*; *GAP43-mCherry* | 6 | Automated |
| *eve-EGFP, GAP43-mCherry* | 3 | Automated, manual correction |
| *wg^CX4^*; *eve-EGFP, GAP43-mCherry* | 3 | Automated, manual correction |

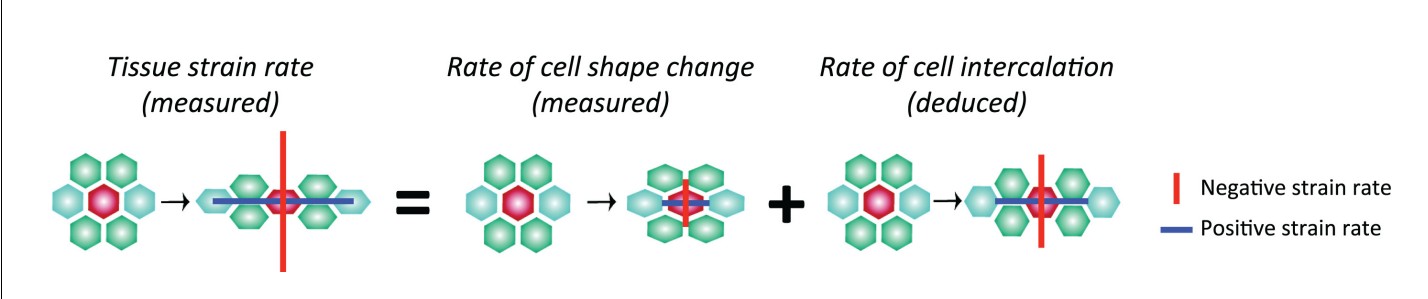

**Figure 8.** Methodology for quantifying tissue strain rates.

between the anterior of the field of view and the ventral mid-line, with positive AP aligned towards the embryonic posterior. The origin of this coordinate system moved with the location of the intersection point, for example if there was any lateral movement of the embryo in AP or if the embryo rolled in DV.

## Domain strain rates

Using the relative movements of cell centroids, local tissue 2D strain (deformation) rates were calculated for small spatio-temporal domains (see *Figure 8* below and *Blanchard et al., 2009*; *Butler et al., 2009*; *Lye et al., 2015*), composed of a focal cell and one corona of neighbouring cells over a 2 min interval (contained within five movie frames). A separate direct measure of 2D cell shape change was calculated by first approximating each cell with its best-fit ellipse, then finding the strain rate tensor that best mapped a cell's elliptical shape to its shape in the subsequent time point. The difference between the local tissue strain rates (calculated above) and the average cell shape strain rate of cells in the same spatio-temporal domain was attributed to cell intercalation. All strain rates were then projected onto the embryonic axes, AP and DV.

## Movie synchronization in time

WT movies were synchronized in time (*Figure 1—figure supplement 1A*) with zero min defined as the last frame in which there was no extension at the posterior edge of the image. This was further refined to the frame before which tissue extension in the AP axis exceeded a proportional rate of 0.01 / min. We confirmed that ectodermal cell division and the timing of the cessation of cell intercalation in different embryo movies were clustered in time as a result (*Figure 1—figure supplement 1B*).

## Cell selection

For all analyses, we included only neurectoderm cells, having classified and excluded all head, mesoderm, mesectoderm, non-neural ectoderm and amnioserosa cells.

## Myosin II quantification

We subtracted the modal pixel intensity as background from raw images in the Myosin II channel at each time point. We set the width of cell-cell interfaces at 3 pixels, a compromise between being wide enough to encompass all interface Myosin II fluorescence, and narrow enough to minimise the inclusion of medial Myosin II. The fluorescence intensity for each cell-cell interface in each movie frame (every 30 s) was calculated as the average intensity of interface pixels.

## Measures of bidirectional and unidirectional polarity

We measured apical cell membrane Myosin II polarity using the Myosin II fluorescence intensities of each cell-cell interface calculated above. We first expressed interface fluorescence intensity around each cell perimeter as a function of angle, from the embryonic posterior (zero) anti-clockwise (*Figure 1—figure supplement 2A*). Treating this intensity signal from 0 – 360 degrees as a periodic repeating signal, we calculated its Fourier decomposition, extracting the amplitude of the period 2

component as the strength of Myosin II bipolarity (equivalent to planar cell polarity), with its phase representing the orientation of cell bipolarity (*Figure 1—figure supplement 2B–D*, red lines). We also extracted the period 1 component as a Myosin unipolarity measure (*Figure 1—figure supplement 2B–D*, cyan lines). The orientations of both uni- and bipolarity distributions for our dataset were strongly and consistently biased towards the AP-axis (*Figure 1C', D'*). However, there was some pollution of the unipolarity signal in the bipolarity signal, with the latter enhanced because of the castellated (discontinuous) nature of the average interface intensity signal (*Figure 1—figure supplement 2A–D*, black lines). We therefore explored further methods to calculate independent uni- and bipolarity quantities. Based on the consistent AP bias to both kinds of polarity, we measured the polarity in the AP axis only. We found that fitting two Gaussians independently, centred on the anterior and posterior sides of each cell works well, and is able to separate combinations of uni- and bipolarity (*Figure 1—figure supplement 2B'-D'*).

We fitted the amplitudes and variances of anterior and posterior Gaussians through minimising the discrepancy between the combined Gaussian signal and the Myosin II signal. The bipolarity signal was taken as two peaks of the amplitude of the smaller of the two Gaussians. Subtracting the bipolarity signal from the combined Gaussians, the remainder is the unipolarity signal. Because overall Myosin II intensity differed between embryos, we normalised the strength of both polarities by dividing the allocated Gaussian amplitude area by the cell's mean perimeter Myosin II signal, so that they would be consistent across embryos.

Finally, we made an adjustment to account for an imaging artefact that results in a domed intensity of Myosin II in all images, with corners less bright than the image centres. The differences in brightness are not an issue per se, since we express polarity amplitudes as a proportion of mean cell perimeter fluorescence, but an artefactual gradient in intensity across a cell will introduce a unipolarity signal. We therefore fitted a smooth to the Myosin II intensity across each image separately (with a kernel size of 1/20$^{th}$ of the image width), calculated the local gradient of this smooth for each cell, and rebalanced the local gradient effect while keeping the mean cell perimeter fluorescence the same.

Using the above methods we produced uni- and bipolarity measures projected along the AP axis for each cell at each time point, that are independent of each other and normalised to control for variation in Myosin II fluorescence.

## Contoured heat maps

Heat maps show time on the y-axis plotted against some measure of AP location on the x-axis, with heat colour representing a third variable. Variation of the third variable was averaged over the DV axis. Heat maps show the mean values of the third variable for each grid square of the plot, the size of which is shown in 'N' heat maps. For example, for *Figure 3B,C*, the 'N' heat map is *Figure 3—figure supplement 1A*, with 80 time bins and 60 AP coordinate bins. White guidelines drawn over contoured heat maps are the average cell trajectories, showing the gross extension of the tissue in the AP axis over time.

## Defining PSB interfaces and cell types

Tissue domains were defined in individual tracked movies using two different techniques, depending on the embryo's genotype. For movies of *sqh$^{AX3}$; sqh-GFP; GAP43-mCherry* embryos, strong PSB enrichments of Sqh-GFP were identified at the end of movies. Groups of cells in between strong Sqh-GFP enrichments were manually selected (each group corresponding to a single parasegment) in a single time point at the end of each movie. Because cells were tracked over time, these classifications of parasegmental group identity could be automatically backtracked to define the same groups of cells at all earlier time points.

For movies of *eve-EGFP, GAP43-mCherry* embryos, the anterior boundaries between parasegments were identified by clear anterior margins of Eve-EGFP positive nuclei. Groups of cells in between successive clear anterior margins of Eve-EGFP positive nuclei were manually selected (each group corresponding to two parasegments) in a single time point at the end of each movie. Groups of cells were again classified at earlier time points by backtracking through movies.

We only used parasegments that were seen throughout each movie, excluding, for example, posterior parasegments that flowed out of the field of view as a result of axis extension. Data used in

**Table 2.** Summary of parasegments analysed for each *sqh^AX3^; sqh-GFP; GAP43-mCherry* movie.

| Movie Identifier | PS4 | PS5 | PS6 | PS7 |
|---|---|---|---|---|
| SG_1 | ✕ | ✓ | ✓ | ✓ |
| SG_2 | ✕ | ✓ | ✓ | ✓ |
| SG_3 | ✓ | ✓ | ✓ | ✕ |
| SG_4 | ✓ | ✓ | ✓ | ✕ |
| SG_5 | ✓ | ✓ | ✓ | ✕ |
| SG_6 | ✓ | ✓ | ✓ | ✓ |

subsequent analyses were from parasegments 4–7 (summarised in *Table 2*), as calculated from the distance along the AP axis of the embryo, and from the timing and location of cell division nests in abdominal parasegments (*Foe, 1989*).

## Quantifying interface co-alignment

Interface orientations, relative to the embryonic axes, were calculated for PSB, -1 and +1 interfaces at all time points in movies from *eve-EGFP, GAP43-mCherry* or *wg^CX4^; eve-EGFP, GAP43-mCherry* and for PSB, S1/2B, S2/3B, +1 and -1 interfaces at all times points in movies from *sqh^AX3^; sqh-GFP, GAP43-mCherry* embryos. All distributions of interface orientations (from 0, parallel to the AP embryonic axis, to 180°) were reflected around 90°, producing distributions from 0°, AP-aligned, to 90°, DV-aligned. As a measure of co-alignment, the proportion of interfaces oriented between 60 and 90° relative to the AP axis was plotted over time, from -20 to 60 min. Cumulative frequencies were calculated for each reflected distribution of interface orientations at 40 min (corresponding to when Myosin II levels were significantly different). Two-sample Kolmogorov-Smirnov tests on the cumulative frequency distributions of interface orientation were used to compare treatments (Prism, GraphPad).

We repeated the analysis, treating the distribution of interface angles from 0° to 180° as a circular distribution, and calculating the parameter of concentration (κ) of the von Mises probability density function. Both the plots of κ versus time and the comparison of distributions at 40 min gave very similar results to the above methods (data not shown).

## Laser ablation of cell-cell interfaces

Junctional laser ablation experiments were carried out in *sqh^AX3^; sqh-GFP; GAP43-mCherry* embryos. PSBs were located by eye by finding i) connected junctions that had the strongest Sqh-GFP intensity and ii) had mirror image Sqh-GFP enrichments the other side of the embryonic midline. We confirmed that the PSB interfaces we selected were significantly more strongly enriched in Myo II than + 1 interfaces by quantifying the mean Sqh-GFP intensity in a line section drawn over the junction at the time point prior to ablation (*Figure 2—figure supplement 1H*). We also confirmed that +1 interface orientations relative to the embryonic midline were more broadly distributed than PSB interface orientations (*Figure 2—figure supplement 1I*). Further quantification showed that interface types did not differ in mean length (*Figure 2—figure supplement 1G*).

Ablations were carried out as described in *Lye et al. (2015)*. 2 to 4 ablations were performed in each embryo and a total of 15 embryos were used. 19 ablations were carried out for both PSB and +1 junctions. A single ablation was performed in each parasegment and all ablations were confined to the Vnd and Ind domains along the DV axis of the embryo. The region of interest selected for ablation was placed over the middle of the chosen junction. 5 images were collected prior to ablation (any longer and the junction would move away from the region of interest due to axis extension movements) and up to at least 30s after ablation.

Line sections were then manually drawn over ablated junctions and the Dynamic Reslice tool in ImageJ was used to produce kymographs. The distances between the two vertices at either end of a junction were measured from 5 time points before ablation until 30s after ablation. Linear regression was performed on the first 5 time points after ablation. The slope of the regressed line was used as

a measure of the vertex recoil velocity. The 'equal slopes' test function in Prism (GraphPad) was used to test for significant differences between slopes and thus difference in recoil velocities.

## Heat map statistics

Pooling the normalised polarity data from 6 $sqh^{AX3}$; *sqh-GFP*; *GAP43-mCherry* embryos, each grid square of the heat maps has a distribution, with number of data points per grid square (*Figure 3—figure supplement 1A*), mean (*Figure 3B,C*) and a confidence interval that we can calculate. We tested whether the mean value in each grid square of contoured heat maps (averaged over the 6 embryos) is significantly different from zero using t-tests. *Figure 3—figure supplement 1B,C* show squares in white that are not different from zero at the 95% two-tailed confidence level. In *Figure 3—figure supplement 1C*, where unipolarity is significantly different from zero, the direction rather than the strength of unipolarity (see *Figure 3C*) is shown.

## Calculation of cells per parasegment and stripe width

For each parasegment, we calculated the average width of the parasegment in AP (psw) and the average width of each cell in that parasegment, also in AP (cw). To give the average number of cells per parasegment width, we divided psw by cw (*Figure 3D*). For the number of cells per stripe width, the numerator was the width of the stripe (*Figure 4D*).

## Assignment of stripe boundaries

We manually defined within-parasegment stripe boundaries, looking for Myosin II accumulation along DV interfaces linked in cable-like structures parallel to PSBs and classifying cells as being in stripe S1, S2 or S3 within each parasegment. We checked our stripe classifications by plotting the locations of stripe boundary (scoring 0) and non-boundary (scoring 1) interfaces against within-para-segment coordinate (*Figure 4C*). The peaks in location of boundary interfaces align very well with the mean location of within-parasegment boundaries S1/2B and S2/3B (black arrows) taken from *Figure 3C* and *Figure 3—figure supplement 1C*.

## Classifying productive neighbour exchanges

We registered neighbour exchange events when a cell-cell interface swapped ownership from one pair of neighbours to an orthogonal pair of neighbours. Most neighbour exchange events were straightforward, with the reducing interface swapping cleanly into a new growing interface. However, some swapped repeatedly before resolving, or did not resolve, or reverted to the original cell connectivity. We therefore set a threshold time window of 5 min over which we ignored repeated neighbour swaps.

## Deviation from Voronoi tessellation

We expected that polarised Myosin II (uni- or bidirectional) at cell junctions would lead to cell shapes that differed from relaxed geometries of a kind that would be expected if, for example, Myosin II was either absent or uniformly distributed. We therefore constructed a measure to quantify the degree of difference from a putative relaxed geometry, both at the scale of cell perimeters and of individual cell-cell interfaces. We first defined relaxed geometries. We chose a Voronoi tessellation, based on cell centroid locations, as a simple first approximation to relaxed geometries. A Voronoi tessellation identifies cell-cell interfaces as the set of points equidistant from two neighbouring cell centroids. Vertices are located where these interfaces from local pairs of cell centroids intersect (*Figure 5D*). The tessellation will stretch with tissue (cell centroid) stretch, so we expected our comparisons to be robust to cell elongation per se.

Using existing cell centroids (centres of mass), we used a Voronoi tessellation to obtain expected vertex locations, cell-cell interface lengths and cell perimeters. We quantified the difference between actual cell perimeters and those based on Voronoi predictions. By definition, as cell shapes become geometrically stressed, cell perimeters will on average become longer than those predicted by the Voronoi tessellation. We subtracted tessellated interface lengths from observed interface lengths to get a measure of geometric stress. A value near zero indicated a relaxed geometry, with increasing deviation from zero indicating increasingly stressed geometries (*Figure 5—figure supplement 1D*). We expected relaxed cell geometries twenty min before the start of GBE, when cells have finished

cellularisation but before gastrulation and before polarised Myosin II expression. Indeed, perimeter stress was low and stable until the start of GBE, when it rose sharply then remained high throughout GBE (*Figure 5E*, black line). Mesoderm invagination no doubt introduces some stress from -5 to 5 min (*Lye et al., 2015*), but the fact that the geometric stress index remained high thereafter shows that this stress is likely to be actively maintained in the germ-band.

Boundary interfaces behaved differently from non-boundary interfaces, with the latter longer than expected (*Figure 5E*). We investigated further, aligning interfaces in time to zero at the point of neighbour exchange. The deviation of boundary interface length increased prior to exchange events, coinciding with a similarly increase in interfacial myosin prior to exchange as interfaces shortened (*Figure 5F*). Upon neighbour exchange, these interfaces became non-boundary interfaces and showed an elongated signature. Overall, these data suggest that the active contraction of boundary interfaces is driving convergence in DV, and that as soon as they become non-boundary interfaces they take on a passive signature.

## Combinatorial receptors patterns

The scoring for each permutation is explained in *Figure 6* and *Figure 6—figure supplement 1*. The code for generating the permutations is given in *Source code 1*.

## Vertex model of axis extension

We used mathematical modelling to investigate the mechanical implications of actomyosin planar polarisation during *Drosophila* axis extension. Vertex models are a particularly successful description of epithelial mechanics that model the polygonal tessellation that cells' adherens junctions form in two dimensions (*Farhadifar et al., 2007*; *Fletcher et al., 2014*; *Honda and Eguchi, 1980*). In such models, the movement of junctional vertices and the rearrangement of cells are governed by the strength of cell-cell adhesion, the contractility of the actomyosin cortex and cell elasticity.

## Governing equations

We describe the epithelial sheet by a set of connected vertices in two dimensions. Assuming that the motion of these vertices is overdamped, the position $r_i(t)$ of vertex $i$ evolves according to the first-order equation of motion

$$\eta \frac{dr_i(t)}{dt} = F_i(t),\qquad(1)$$

where $F_i(t)$ denotes the total force acting on vertex $i$ at time $t$ and $\eta$ denotes the common drag coefficient. We specify the forces acting on vertices through a 'free energy' function $U$, for which

$$F_i = -\frac{\partial U}{\partial r_i}.\qquad(2)$$

Our choice of $U$ is based on that proposed in *Farhadifar et al. (2007)* and is given by (see *Figure 7A*):

$$U = \sum_\alpha \frac{K}{2}(A_\alpha - A_0)^2 + \sum_\alpha \frac{\Gamma}{2}P_\alpha^2 + (i,j)\sum f(l_{ij}).\qquad(3)$$

The first term in this free energy function describes an area elasticity with common elastic coefficient $K$, for which $A_\alpha$ is the area of cell $\alpha$ and $A_0$ is a common 'target' area, and the sum runs over all cells at time $t$. The second term describes the contractility of the cell perimeter $P_\alpha$ by a common coefficient $\Gamma$, with the sum again running over all cells at time $t$. The third term represents 'line tensions' at cell-cell interfaces, where $l_{ij}$ denotes the length of the edge shared by vertices $i$ and $j$ and the sum runs over the set of cell-cell interfaces at time $t$. Line tensions can be reduced by increasing cell-cell adhesion or reducing actin-myosin contractility. The precise functional form of this line tension energy term varies across our simulations.

In addition to these equations of motion for cell vertices, we need to ensure that cells are always non-intersecting and to allow cells to form and break bonds. This is achieved through an elementary operation called edge rearrangement (a T1 transition or swap), which corresponds biologically to cell intercalation. Mathematically, such arrangements are necessary in the vertex model due to the

finite forces acting on a cell's vertices arbitrarily far from equilibrium. We implement a T1 swap whenever two vertices $i$ and $j$ are located less than a minimum threshold distance $d_{min}$ apart (taken to be much smaller than a typical cell diameter). In this case, the two vertices are moved orthogonally to a distance $pd_{min}$ apart and the local topology of the cell sheet is modified such that they no longer share an edge.

The configuration of the cell sheet is updated using the following algorithm. Prior to numerical solution, we non-dimensionalize the model, following previous implementations (*Farhadifar et al., 2007*; *Kursawe et al., 2015*) by rescaling all lengths with $\sqrt{A_0}$ and all times with $\eta/KL^2$; thus, all presented model results are non-dimensional. Starting from an initial configuration $r_i(0)$, we update the state of the system until time $T$ over discrete time steps $\Delta t$. At each time step we: implement any required T1 swaps; compute the forces $F_i$ on each vertex from the free energy $U$; solve the equation of motion for each vertex over the time step numerically, using an explicit Euler method; and finally update the positions of all vertices simultaneously. We implement this model in Chaste (*Fletcher et al., 2013*), an open source C++ library that allows for the simulation of vertex models. The code is given in the file *Source code 1*.

## Simulations

We consider several alternative model simulations of axis extension, which differ only in the hypothesised dependence of the line-tension energy described above on the length and type of cell-cell interfaces. In each simulation, we model the movement, shape change and neighbour exchange of a small tissue that is initially comprised of 20 rows and 14 columns of hexagonal cells. Prior to the start of each simulation, we simulate the evolution of the tissue to mechanical equilibrium under the assumption that the line-tension energy varies linearly with interface length, $f(l_{ij}) = \Lambda_{ij}l_{ij}$, with the same (constant) coefficient for every interface, $\Lambda_{ij} = \Lambda_{int}$. This avoids compounding the later dynamics by artefacts associated with starting the tissue from a non-equilibrium cell size. The value of $\Lambda_{int}$ and all other parameters used in the simulations described below are provided in *Table 3*. We then simulate the tissue until time $T$ under a different hypothesised dependence of the line-tension energy, as described below. In each simulation, we introduce four distinct stripes of cell identities within each parasegment (*Figure 7B*). Note that both stripes 3 and 4 are initially discontinuous, reflecting our in vivo finding that these two stripes have a combined initial AP width of 1.5 cells (S3 in *Figure 4D*).

## Simulation 1 (no feedback or supercontractility)

Here, we follow (*Farhadifar et al., 2007*) in setting the line tension energy to vary linearly with the length of a cell-cell interface:

$$f(l_{ij}) = \Lambda_{ij}l_{ij}. \tag{4}$$

**Table 3.** List of parameters and their values used in simulations.

| Parameter | Description | Value | Simulations |
| --- | --- | --- | --- |
| $\eta$ | Drag coefficient | 1.0 | All |
| $T$ | Simulation end time | 500 | All |
| $\Delta t$ | Time step | 0.001 | All |
| $d_{min}$ | T1 swap threshold | 0.01 | All |
| $p$ | T1 swap distance multiplier | 1.5 | All |
| $K$ | Elastic coefficient | 1 | All |
| $A_0$ | Cell target area | 1 | All |
| $\Gamma$ | Contractility coefficient | 0.04 | All |
| $\Lambda_{int}$ | Line-tension coefficient for non-boundary (or tissue-boundary) interfaces | 0.05 | 1-3 |
| $\Lambda_{bdy}$ | Line-tension coefficient for (stripe-) boundary interfaces | $2\Lambda_{int}$ | 2-3 |
| $\Lambda_{sup}$ | Line-tension coefficient for super-contractile (stripe-) boundary interfaces | $8\Lambda_{int}$ | 4 |

In our model, the line-tension coefficient $\Lambda_{ij}$ takes one of two values, depending on the type of interface. If the interface is shared by two cells of the same stripe identity (a non-boundary interface), or it is contained in a single cell (a tissue-boundary interface), then we set $\Lambda_{ij} = \Lambda_{int}$. If the interface is shared by two cells of different stripes identities (a stripe-boundary interface), then we set $\Lambda_{ij} = \Lambda_{bdy}$, where $\Lambda_{bdy} > \Lambda_{int}$ and thus boundary interfaces are more contractile than non-boundary interfaces (*Figure 7—figure supplement 1A*):

$$f\left(l_{ij}\right) = \begin{cases} \Lambda_{int}l_{ij}, & \text{for a non-boundary or tissue-boundary interface,} \\ \Lambda_{bdy}l_{ij}, & \text{for a stripe-boundary interface.} \end{cases} \tag{5}$$

In this simulation, we find that cells are unable to execute neighbour exchanges and hence axis extension is not achieved (*Figure 7—figure supplement 1B*).

## Simulation 2 (feedback based on individual boundary interfaces)

In our next model, we consider a nonlinear dependence of the line-tension energy $f\left(l_{ij}\right)$ on the cell-cell interface length. Here, we wish to study the effect of including a feedback or runaway component, in which shorter stripe-boundary interfaces become enriched in Myosin II and thus more contractile, on the axis extension process. To this end, we choose the functional form (see *Figure 7—figure supplement 1A*):

$$f\left(l_{ij}\right) = \begin{cases} \Lambda_{int}l_{ij}, & \text{for a non-boundary or tissue-boundary interface,} \\ \Lambda_{bdy}\log l_{ij}, & \text{for a stripe-boundary interface.} \end{cases} \tag{6}$$

In this simulation, we find that while some cells exchange neighbours, most 4-way junctions do not resolve (*Figure 7C*).

## Simulation 3 (feedback based on contiguous boundary interfaces)

To address the resolution of 4-way junctions encountered in Simulation 2, we next consider a more complex model of line tension, where now the value of the coefficient $\Lambda_{ij}$ is computed as follows for boundary interfaces. For each of the two cells sharing the boundary interface, we sum the lengths of the (contiguous) boundary interfaces shared by the cell, including the boundary interface of interest. Having computed this number for each of the two cells, we then compute the smaller of these two numbers, which we denote by $L_{i,m}$, the indices reflecting the variable number of contiguous boundary interfaces (*Figure 7E*). The line-tension coefficient then takes the form:

$$f\left(l_{ij}\right) = \begin{cases} \Lambda_{int}l_{ij}, & \text{for a non-boundary or tissue-boundary interface,} \\ \Lambda_{bdy}\log L_{i,m}, & \text{for a stripe-boundary interface.} \end{cases} \tag{7}$$

This resolves the 4-way junction issue encountered in simulation 2 because cells are able to shear along boundaries (*Figure 7F*). However, the initially discontinuous stripes 3 and 4 remain in clumps, unable to repair their stripe continuity.

## Simulation 4 (inclusion of super-contractility)

Our next model builds on Simulation 3 to include super-contractility, in which the line-tension coefficient $\Lambda_{ij}$ now takes a different value for boundary interfaces between stripes whose identities differ by 2 (for example, an interface between a cell belonging to stripe 2 and a cell belonging to stripe 4) to those between stripes whose identities differ by 1. We denote these values by $\Lambda_{sup}$ and $\Lambda_{bdy}$, respectively, where $\Lambda_{sup} > \Lambda_{bdy}$ to reflect our hypothesis that mismatched or skipped identity boundary interfaces are more contractile than other boundary interfaces:

$$f\left(l_{ij}\right) = \begin{cases} \Lambda_{int}\, l_{ij}, & \text{for a non-boundary or tissue-boundary interface,} \\ \Lambda_{bdy}\log L_{i,m}, & \text{for a stripe-boundary interface,} \\ \Lambda_{sup}\log L_{i,m}, & \text{for a skipped stripe-boundary interface.} \end{cases} \tag{8}$$

In this simulation stripes 3 and 4 repair their continuity and the patterns of cell and interface behaviours qualitatively mimic in vivo data (*Figure 7G* and compare *Figure 7—figure supplement 1C* with *Figure 5I*).

## Acknowledgements

We thank Jocelyn Etienne, Daniel St Johnston, David Strutt, Rob White and members of Bénédicte Sanson's lab for critical reading of the manuscript and discussions, and Erik Clark for discussions.

## Additional information

### Funding

| Funder | Grant reference number | Author |
| --- | --- | --- |
| Wellcome Trust Graduate Studentship | 089614/Z/09/Z | Robert J Tetley |
| Biotechnology and Biological Sciences Research Council Standard Grant | BB/J010278/1 | Guy B Blanchard Richard J Adams Bénédicte Sanson |
| Wellcome Trust Investigator Award | 099234/Z/12/Z | Bénédicte Sanson |

The funders had no role in study design, data collection and interpretation, or the decision to submit the work for publication.

### Author contributions

RJT, Conceived the project; Performed the experiments; Analysed and interpreted the data; Performed the statistical analysis; GBB, Conceived the project; Analysed and interpreted the data; Developed and updated the 'otracks' software and associated statistical tools; Provided expertise on cell tracking and statistics; Performed the statistical analysis; Developed new computational tools for the analysis of Myosin II polarity and interface behaviours; Developed receptor permutation model; Applied the vertex model to biological problem; AGF, Developed the vertex model; Applied the vertex model to biological problem; RJA, Developed and updated the 'otracks' software and associated statistical tools; Provided expertise on cell tracking and statistics; BS, Conceived and supervised the project; Analysed and interpreted the data; Wrote the manuscript with input from all authors

### Author ORCIDs

Guy B Blanchard, http://orcid.org/0000-0002-3689-0522
Bénédicte Sanson, http://orcid.org/0000-0002-2782-4195

## Additional files

### Supplementary files

• Source code 1. Code for receptor permutations (*Figure 6*) and vertex model (*Figure 7*), as cited in the Materials and methods.

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
