## [Decision Letter]

Thank you for submitting your work entitled "Unipolar distributions of junctional Myosin II identify stripe boundaries driving cell intercalation in axis extension" for consideration by *eLife*. Your article has been reviewed by three peer reviewers, and the evaluation has been overseen by a Reviewing Editor and K VijayRaghavan as the Senior Editor.

The reviewers have discussed the reviews with one another and the Reviewing Editor has drafted this decision to help you prepare a revised submission.

This study investigates the cellular basis of convergence extension in the fly embryo. It describes with unprecedented resolution the stereotyped changes in the planar distribution of non-muscular Myosin II (MyoII) in wild-type embryos. To do so, the authors combined live imaging with computational methods to precisely quantify the dynamics and the spatial patterns of MyoII planar polarization. The authors showed that the initial bipolar distribution of MyoII is most pronounced at parasegmental boundaries and that this initial bipolar distribution is rapidly followed by its unipolar accumulation at stereotyped positions relative to parasegmental boundaries. This shift from bipolar to unipolar correlates temporally with cell intercalation. Moreover, the authors show that MyoII accumulation better correlated with the juxtaposition of cells of different identities than with the orientation of the cell-cell interfaces relative to the body axis. These findings led the authors to revise the current Toll receptor model to explain how different cell identities generate the planar polarization of MyoII, hence germ-band extension.

The paper is very well-written and the quantifications are extremely well-done and precise. The consistent quantification of the process over 80 minutes of development is in itself a significant achievement and provides with a nice picture of how myosin redistributes during germ-band elongation. Taken together, the paper is a step forward in our understanding of the very subtle mechanisms that control convergent-extension in *Drosophila*.

However, several points need to be addressed before the paper can be considered for publication in *eLife*.

Essential revisions:

The authors have performed an extensive analysis of 6 embryos during GBE. However, the quantitative analysis is difficult to follow. For example, it is not clear what is displayed in Figure 1 and G. How is "pp" defined? What defines the position along the AP axis? Cell center of mass? Is it a projection of all cells along the DV axis?

In Figure 4, the authors plot the average Myosin 2 intensity over time for different junctions within a parasegment. The differences in myosin levels between the S1/2B and +1/-1 interfaces are relatively small. Particularly between 20 minutes and 30 minutes, where at the same time Figure 3 show a clear transition. This suggests that the effect quantified on Figure 3 are rather small, and additional evidence for its mechanical influence should be added. Estimation of tension in these different junctions (e.g. by laser ablation) would bring additional support to a differential behavior of these junctions.

The quantification of T1 transitions, and in particular the data presented in Figure 4—figure supplement 4I, are not entirely convincing (are these data pooled from several embryos? why not present means with error bars?); could the authors reinforce these data, e.g. by investigating correlations between myosin enrichment and T1 transitions?

Figure 5 suggests the authors are proposing a mechanism that would simultaneously explain myosin accumulation at parasegments boundaries and give robustness to variation in the number of cells across the AP direction in each parasegment, but the latter is unclear to me. Is it clear that the rows 3 and 4 stay separated at the end of germ-band elongation? And if so, can the authors clarify the mechanism they envision that would provide with this robustness?

Finally, the entire study is performed on WT embryos and the conclusions are based on correlations. The presented data support a model of local cell-cell interactions driving inhomogeneity in tissue mechanics that will drive GBE. Performing the same quantitative analysis on a perturbed case (e.g. Toll receptor mutant) would significantly strengthen the study. We are aware that such an investigation might be beyond the scope of the current paper, but such data would strongly increase the impact of the mechanism proposed.

[Editors' note: further revisions were requested prior to acceptance, as described below.]

Thank you for resubmitting your work entitled "Unipolar distributions of junctional Myosin II identify cell stripe boundaries that drive cell intercalation throughout *Drosophila* axis extension" for further consideration at *eLife*. Your revised article has been favorably evaluated by K VijayRaghavan (Senior editor), a Reviewing editor, and three reviewers.

The manuscript has been very much improved but a few remaining issues raised by two of the reviewers need to be addressed before acceptance. These are mostly clarifications concerning the model, as outlined below:

*Reviewer #2:*

The authors have added vertex model simulations which are interesting. They have introduced a term for the line tension that depends on the total interfacial length at the boundary, which appears to be required to avoid the system being stuck in 4-fold vertices. It seems to me that it is a bit unclear why this term has such an effect – the schematic in Figure 7' is supposed to illustrate this but it is hard to understand. Also, can the authors briefly discuss which biological process could give rise to such a term?

In Figure 7 and its legend there is a confusing denomination of the term *f(l_ij_)* in the energy as being called "line tension". Strictly speaking the line tension is the derivative of the energy with respect to the interface length, Λ*_ij_*. In the main text and Methods the authors are referring to this term as a "line tension energy", which sounds more appropriate. This also implies than when f is a linear function of Λ*_ij_*, this correspond to a condition where the line tension is constant.

In the model, the parameter Λ*_bdy_* is defined with a different dimension in Eqs. 5-6 and in the second equation in the subsection “Simulation 1 (no feedback or supercontractility)” (there is a length scale of difference). I assume that the authors take as a reference length scale the square root of the cell target area, but I think this should be clarified.

In Video 5 showing vertex model simulations, there appear to be edges that randomly appear and disappear in some frames – is that an artefact of the simulation?

*Reviewer #3:*

1) The authors have added a vertex model to support their analysis. I appreciate the add on of this model. Even if the different description of the line tension in the boundaries and supercontractile junctions seems adhoc to me, this is fine as a first description. However, the parameters used for the line tension indicate a factor 2 and 8 respectively between non boundary, boundary and "supercontractile boundary". The authors should discuss whether these parameters seem reasonable. To me, the factor 2 could be consistent with the data in Figure 2, but the factor 8 seems exaggerated.

2) In Figure 6, for clarity I would add a drawing of the corresponding segments when 1 cell and 2 cells are missing in the supplementary information.

3) In Figure 2, The Figure 2 could be placed in supplementary information and the Figure 2 scale bars are missing as well as a scale for the time.

---

## [Author Response]

*Essential revisions:*

*The authors have performed an extensive analysis of 6 embryos during GBE. However, the quantitative analysis is difficult to follow. For example, it is not clear what is displayed in Figure 1 and G. How is "pp" defined? What defines the position along the AP axis? Cell center of mass? Is it a projection of all cells along the DV axis?*

To improve clarity, we have extensively revised Figure 1 and associated figure supplements. We have brought elements from the supplement figures to explain more step by step our quantification methods. Note that we have used two distinct methods for Myosin II polarity quantification, called “Fourier” and “Gaussian”, which we explain in Figure 1—figure supplement 2, in addition to in the Methods. Specifically:

In Figure 1, we have added the new panels C, C’ and D, D’. C and D show an example of how Myosin II bipolarity (C) and unipolarity (D) is calculated for each cell, using our first Fourier method; we show two movie frames where bipolar and unipolar vectors are drawn from every cell centroid and point towards either the two sides (bipolar) or one side (unipolar) of the cells which are/is most enriched in Myosin II. We then show in C’ and D’ how these vectors are well oriented along the AP axis, averaging for the 6 videos (we use the ventral midline, VML, of the embryo as our reference for AP axis orientation in the videos). This strong AP orientation of the Myosin II bidirectional and unidirectional polarities justifies our use of a second Gaussian fitting method, with which we quantify the strength of polarity only along the AP axis (C’’ and D’’ and E and F and other similar data).

We have added a couple of movie frames again in Figure 1, to explain better the spatiotemporal maps. One movie frame corresponds to the start of GBE (time zero) and shows all of the tracked cells that are considered in the analysis. It clarifies for example, that the mesoderm and mesectoderm cells present on either side of the invaginating ventral furrow (VF) at this stage are excluded from the analysis. Also, the germband cells on the posterior side of the field of view are not analysed, because these disappear from the field of view as the tissue extends. A second movie frame shows the embryo at 50 minutes. Most of the cells that were excluded from the analysis at time zero are now gone (mesoderm cells have invaginated and posterior germband cells have moved out of the field of view). The only remaining cells that do not make it in the analysis are the mesectoderm cells at the ventral midline (VML) and the dorsal ectoderm cells, which are just starting to appear in the field of view.

The spatiotemporal maps in Figure E and F show the data as a function of GBE time (y-axis) and position along AP in microns from the anterior edge of the field of view (x-axis). The data is averaged along DV for every position (bin) along AP. The video frames now help to represent this data averaging in DV and we have also added a sentence in the Results to clarify this point: “Note that in these maps, the data for all cells along the dorsoventral (DV) axis, within an AP bin of defined width, are averaged.”

The unitless values we use for our quantification of Myosin II bi- and unidirectional polarities are defined in the legends of Figure 1: “The amplitude of Myosin II bipolarity is expressed as a proportion (Abbreviated as pp in all figures) of the mean Myosin II intensity around the perimeter of each cell.” The amplitude calculations are explained visually in Figure 1—figure supplement 2, in addition to in the Methods.

To improve clarity, we have also made changes in the figure supplements for Figure 1. We now parcel the information in three distinct supplements:

Figure 1—figure supplement 1 gives information about how we synchronized the 6 *sqh^AX3^; sqh-GFP; GAP43-mCherry* movies. We have also added a new panel (C) giving the total number of cells analysed as a function of GBE time, for the 6 synchronised videos. At time zero, we analyse about 800 cells, and this increases to about 2000 cells per time point between 25 to 40 minutes of GBE.

Figure 1—figure supplement 2 explains visually the two distinct methods we used for calculating Myosin II polarities. We have improved the clarity of the figure and legends, and added two new panels A and A’ to explain how Myosin II enrichments are measured for each cell analysed.

Figure 1—figure supplement 3 shows the individual spatiotemporal plots for all 6 *sqh^AX3^; sqh-GFP; GAP43-mCherry* movies analysed. Note that all plots used the Gaussian method for calculating polarities, except for panel A, where we show an example of a plot done using the “Fourier” method for Video 4 (The corresponding Gaussian plots for this movie are given in Figure 1).

*In Figure 4, the authors plot the average Myosin 2 intensity over time for different junctions within a parasegment. The differences in myosin levels between the S1/2B and +1/-1 interfaces are relatively small. Particularly between 20 minutes and 30 minutes, where at the same time Figure 3 show a clear transition. This suggests that the effect quantified on Figure 3 are rather small, and additional evidence for its mechanical influence should be added. Estimation of tension in these different junctions (e.g. by laser ablation) would bring additional support to a differential behavior of these junctions.*

We did provide an estimation of tension for all boundaries in the submitted manuscript, based on the assumption that if interfacial tension is high along a line of connected interfaces, these interfaces should straighten. In a previous study, we used a basic “index of straightness” to quantify this (Monier et al. (2010). Nature Cell Biology, 12: 60–65. http://doi.org/doi:10.1038/ncb2005). In the present study, we calculated instead how clustered the cell-cell interface orientations were relative to the DV axis (perpendicular to the AP-oriented ventral midline) for a given boundary, using -1 (one cell diameter away to the anterior) and +1 (posterior) interfaces controls. If the boundaries are straight, the orientations should be clustered around the orientation of the DV axis (90 degrees relative to AP). We compared the degree of clustering using circular statistics with highest kappa values indicating strongest clustering of interface orientations. These graphs were given in the previous version of the manuscript and showed that PSBs, S1/2B and S2/3B boundaries have a higher kappa than flanking (+1, -1) interfaces.

To make these data more accessible, we are now providing a slightly more intuitive measure. In addition, we provide a laser ablation experiment to validate this analysis and to provide additional evidence (see below). Instead of providing kappa values, we now provide graphs where we plot the proportion (pp) of interfaces oriented between 60 to 90 degrees relative to the AP axis (so DV-oriented) as a function of time, for all boundaries and the control flanking interfaces. These graphs are now provided in new panels in Figure 2, Figure 2—figure supplement 1 and Figure 4. We have grouped the graphs to make them easier to compare. Note that this includes a new fifth graph, examining PSB interface alignment in a *wingless* mutant (see response to major point 5 below). A statistical comparison is made at 40 minutes for each graph, and the cumulative interface orientations at this time point is shown in 5 new panels in Figure 2—figure supplement 1 and Figure 4. Together, these data show that boundary interfaces at PSBs, S1/2B and S2/3B align more than flanking interfaces. The curves are very similar to those with the kappa values (which we don’t show anymore, but see initial submission), and a brief description of both measures is given in the Methods.

We think that the above measures give a very robust assessment of boundary straightness, sampling hundreds of cell-cell interfaces for each boundary class and indicating higher tension at our 3 boundaries per parasegment. In addition, to respond to the reviewers’ point, we have performed laser ablations on the PSBs and control +1 interfaces to validate that our alignment measure is indeed an indication of interfacial tension. In *sqh^AX3^; sqh-GFP; GAP43-mCherry* embryos, we identified PSB interfaces by their enrichment in Myosin II at about 40 minutes after GBE onset and ablated these interfaces and +1 interfaces as a control (N=19 for each class). We measured the distance between recoiling cell vertices as a function of time and found that PSBs interfaces recoil faster than +1 interfaces. The estimated recoil velocity differs by a factor of about 2 between experiment and control. This shows that PSB interfaces have higher interfacial tensions than +1 interfaces and corroborates our alignment analysis. We infer from this validation that S1/2 and S2/3 boundaries have also higher interfacial tension than flanking interfaces. Note that we attempted initially to perform the ablations in *eve-EGFP, GAP43-mCherry* embryos, where PSBs would have been labelled without relying on visual inspection of Myosin II enrichment. However, it proved impossible to do these ablations, as not only GAP43-Cherry gave a low signal with the two-photon we were using, but also caused sample overheating. We also considered performing ablations of S1/2B and S2/3B interfaces, in addition to the PSBs (using *sqh^AX3^; sqh-GFP; GAP43-mCherry* embryos), but renounced because of the number of ablations we would have to perform to have a conclusive result, which would have taken us beyond a reasonable revision time. We feel that the laser ablation experiment we provide on the PSB validates adequately our more sensitive measures of interface alignment (which consider hundreds of interfaces for a given class) that we have performed on all three classes of boundaries.

*The quantification of T1 transitions, and in particular the data presented in Figure 4—figure supplement 4I, are not entirely convincing (are these data pooled from several embryos? why not present means with error bars?); could the authors reinforce these data, e.g. by investigating correlations between myosin enrichment and T1 transitions?*

We have revised extensively the quantifications of interface behaviours (performed on pooled data for the 6 *sqh^AX3^; sqh-GFP; GAP43-mCherry* embryos). Now, these analyses have their own new Figure 5 and Figure 5—figure supplement 1:

Figure 5 and Figure 5—figure supplement 1’ investigates the relationship between Myosin II enrichment, interface orientation and interface class (either boundary or non-boundary). This shows that the strongest relationship is between Myosin II enrichment and the boundary class of interfaces, rather than orientation.

Figure 5 validates our method for tracking T1 transitions by showing that Myosin II enrichment increases during interface shortening prior to a T1 swap, as found in previous studies.

Figure 5 and Figure 5—figure supplement 1 compare interface length with a theoretical length given by a Voronoi tessellation. This shows that boundary interfaces are on average shorter than this theoretical length, whereas non-boundary interfaces are on average longer. We infer from this that boundary interfaces are under greater relative tension than non-boundary interfaces, suggesting that the boundary interfaces are those actively shortening. Consistent with this, boundary interfaces are shorter in average compared to the Voronoi theoretical length, 15 mins prior to T1 swaps (Figure 5).

We developed a new analysis to examine the fate of all interfaces in the course of the movies, provided in the new panels G-K in Figure 5 and also in Figure 5—figure supplement 1. We identify 4 main classes of interface behaviours, which is described in the following paragraph in the Results:

“Next, we examined the behaviour of all interfaces during GBE for S1 and S2 (Figure 5). We identify four main interface behaviours. […] We conclude that the variable number of cells per parasegment along AP causes a faster intercalation rate in the posterior part of the parasegment compared to the anterior part (Figure 5—figure supplement 1).”

The original graph in Figure 4—figure supplement 4I has been replaced by a new one in Figure 5—figure supplement 1 (see F’ for statistics). We confirm that this is for data pooled from the 6 *sqh^AX3^; sqh-GFP; GAP43-mCherry* embryos. This graph shows that the number of T1 transitions correlates with the position of the three main boundaries identified as being enriched in Myosin II; PSBs, S1/2B and S2/3B. We can also find a lesser peak within stripe 3, which we interpret as corresponding to a less prominent or more variably located boundary between cell identities 3 and 4.

*Figure 5 suggests the authors are proposing a mechanism that would simultaneously explain myosin accumulation at parasegments boundaries and give robustness to variation in the number of cells across the AP direction in each parasegment, but the latter is unclear to me. Is it clear that the rows 3 and 4 stay separated at the end of germ-band elongation? And if so, can the authors clarify the mechanism they envision that would provide with this robustness?*

The fact that peaks in both the number of T1 transitions and Myosin II fluorescence intensity are found in the middle of stripe 3 (see above and Figure 5—figure supplement 1) provides evidence that cell identities 3 and 4 remain separated during GBE. We also find that the rate of intercalation convergence along the DV axis (calculated as in our previous studies Blanchard et al. (2009). Nature Methods, *6*(6), 458–464. http://doi.org/10.1038/nmeth.1327 and Butler et al. (2009). Nature Cell Biology, 11(7), 859–864. http://doi.org/10.1038/ncb1894), is higher in stripe 3 compared to stripe 2 or 1 (Figure 5—figure supplement 1). These two independent analyses corroborate each other and suggest that stripe 3 does in fact contain two separate stripes. The variable initial width of stripe 3 makes the location of the cell identities 3/4 boundary variable in space and time, which is why this boundary does not emerge from the Myosin II unipolarity analysis, and why we cannot reliably locate it manually.

We now test the mechanisms that could generate our observed patterns formally in a new vertex model provided in Figure 7 and Figure 7—figure supplement 1. We show in simulation 4 that cell identities 3 and 4 remain separated during axis extension (Figure 7 and Video 5).

Finally, the entire study is performed on WT embryos and the conclusions are based on correlations. The presented data support a model of local cell-cell interactions driving inhomogeneity in tissue mechanics that will drive GBE. Performing the same quantitative analysis on a perturbed case (e.g. Toll receptor mutant) would significantly strengthen the study. We are aware that such an investigation might be beyond the scope of the current paper, but such data would strongly increase the impact of the mechanism proposed.

We are planning to repeat these analyses in Toll-like receptor mutants, although this is a significant investment in time and will require its own study. One challenge with doing this is that we have to combine a mutant background with our *sqh^AX3^; sqh-GFP; GAP43-mCherry* strain. Making a new fly strain containing these 3 components (*sqh^AX3^, sqh-GFP* and *GAP43-mCherry*) along with a mutant is time-consuming and might not necessarily be viable (we have had problems establishing these combinations with segmentation mutants). A more promising avenue is to inject dsRNA against the gene of interest directly in the *sqh^AX3^; sqh-GFP; GAP43-mCherry* strain. This however will mean dealing with variation in the degree of gene inactivation inherent to RNA interference and will probably require sampling more movies, which is again very time consuming. Because of all this, we feel we cannot provide this experiment within a reasonable timeframe for a revision.

Nevertheless, to take into account the reviewers’ comment, we provide two new sets of data, which test our hypotheses. One is a perturbation using a *wingless* mutant, the second a series of simulations using a vertex-based model.

We first examine the interface alignment and Myosin II enrichment at PSBs in a *wingless* mutant. This experiment was motivated by the fact that we found in a previous study that Wingless signalling is required for Myosin II enrichment and boundary straightening after the end of germband extension, at stage 10 (Monier et al. (2010) Nature Cell Biology, 12(1), 60–65. http://doi.org/doi:10.1038/ncb2005). Wingless is expressed in a stripe of cells just anterior to the PSB. Because Wingless expression is detectable from gastrulation onwards, we wondered if it was also required for PSB function during germband extension. Our new data shows that it is not required, suggesting, remarkably, that all aspects of PSB behaviour during GBE is under pair-rule control. We provide two distinct experiments:

We measured interface alignment at PSB interfaces and flanking interfaces in *wg^CX4^; eve-EGFP, GAP43-mCherry* embryos (data pooled for 3 movies, see Figure 2 and Figure 2—figure supplement 1). In these embryos the PSBs interfaces align strongly along DV compared to control interfaces, in a similar manner to our wildtype control embryos (*eve-EGFP, GAP43-mCherry*, Figure 2 and Figure 2—figure supplement 1).

We quantified Myosin II in fixed *wingless* mutant and wildtype embryos, during germband extension (stage 8) and later (stage 9-11). Consistent with the above experiments and with the results from Monier et al. (2010), we find that whereas Myosin II enrichment decreases in *wingless* mutants after the end of GBE, it is not significantly changed during GBE (Figure 2 and Figure 2—figure supplement 1”’).

The second test of the hypotheses put forward in our paper is a new vertex-based model. This is described in a new section at the end of the Results and in Figure 7 and Figure 7—figure supplement 1. We find that we can recapitulate in silico the cell behaviours that we postulate happen in vivo based on our analysis of Myosin II enrichment and interface behaviour.

[Editors' note: further revisions were requested prior to acceptance, as described below.]

The manuscript has been very much improved but a few remaining issues raised by two of the reviewers need to be addressed before acceptance. These are mostly clarifications concerning the model, as outlined below:

Reviewer #2:

The authors have added vertex model simulations which are interesting. They have introduced a term for the line tension that depends on the total interfacial length at the boundary, which appears to be required to avoid the system being stuck in 4-fold vertices. It seems to me that it is a bit unclear why this term has such an effect – the schematic in Figure 7' is supposed to illustrate this but it is hard to understand. Also, can the authors briefly discuss which biological process could give rise to such a term?

To clarify this, we have replaced Figure 7’ by new panels 7D-E” showing what happens with and without the use of ‘contiguous boundary interfaces’. 7D-D’ shows that in Simulation 2, a subset of interfaces shorten rapidly, but because these are trapped at the boundary, they cannot elongate and this generates stable 4-fold vertices. 7E-E’’ shows how Simulation 3 solves this by considering for a given cell, all boundary interfaces as a contiguous interface. The subset of boundary interfaces that shorten can now become non-boundary interfaces and elongate (7E”). Along with changes to the legend and to the accompanying text, we hope this has clarified why Simulation 2 fails and how the contiguous interface tension rescues this in Simulation 3.

One limitation of vertex models is that apposed cortexes are modeled as a single membrane, and in essence the change between Simulation 2 and 3 attempts to go round this limitation. So we think that in biological terms, what we are attempting to model is a situation where cells behave independently on either side of a boundary. For example, junctions could slide independently of each other on either side of the boundary. This is possible in vivo because a boundary is made of two cell-cell interfaces, and each of these cell interfaces could elongate or shorten independently. This could conceivably happen if the two cell cortices on either side of a boundary have different contractile forces. In addition to junctional sliding, cell-cell sliding could occur along the boundary (simple shear), for example if adhesion is decreased there.

We have added a short paragraph to the Discussion to summarise our model findings and the biological implications.

In Figure 7 and its legend there is a confusing denomination of the term f(l_ij_) in the energy as being called "line tension". Strictly speaking the line tension is the derivative of the energy with respect to the interface length, Λ_ij_. In the main text and Methods the authors are referring to this term as a "line tension energy", which sounds more appropriate. This also implies than when f is a linear function of Λ_ij_, this correspond to a condition where the line tension is constant.

We agree, and have changed "line tension" to "line tension energy".

In the model, the parameter Λ_bdy_ is defined with a different dimension in Eqs. 5-6 and in the second equation in the subsection “Simulation 1 (no feedback or supercontractility)” (there is a length scale of difference). I assume that the authors take as a reference length scale the square root of the cell target area, but I think this should be clarified.

Thank you for pointing out this lack of clarity. Prior to numerical simulation, we have implicitly non-dimensionalized the vertex model in a similar manner to Farhadifar et al., (2007) and Kursawe et al. (2015), by rescaling all lengths by l= A0, where A0 is the target cell area, and rescaling all times with τ = η/KL2, where η is the drag coefficient and K is the elastic coefficient. Thus, all presented model results are non-dimensional. We have clarified this point with an additional sentence in the subsection “Governing equations”.

*In Video 5 showing vertex model simulations, there appear to be edges that randomly appear and disappear in some frames – is that an artefact of the simulation?*

This is a known artefact of the visualization tool (Paraview) only, and is not present in the output of our model. We are currently looking into regenerating Video 5 using an alternative visualization tool, and will upload the movie once this is done, but in the meantime we would prefer not to hold up progress with the manuscript and release of the manuscript PDF.

Reviewer #3:

1) The authors have added a vertex model to support their analysis. I appreciate the add on of this model. Even if the different description of the line tension in the boundaries and supercontractile junctions seems adhoc to me, this is fine as a first description. However, the parameters used for the line tension indicate a factor 2 and 8 respectively between non boundary, boundary and "supercontractile boundary". The authors should discuss whether these parameters seem reasonable. To me, the factor 2 could be consistent with the data in Figure 2, but the factor 8 seems exaggerated.

Our vertex-based model is a first exploration of our hypothesized mechanisms of germ-band extension, based on the new data we present. We did not set out expecting to mimic the real data quantitatively. The simulation approach is therefore still phenomenological, with care needed when attempting to map a particular parameter in the model to a particular biological mechanism. In this context therefore, we do not think that a factor of 8 is problematic – the important point is that these supercontractile interfaces have stronger contraction than other boundary interfaces. A discrepancy between the ratios of tension needed for a successful simulation of boundary behaviour and the ratios estimated in vivo by laser ablation has been noted by (Landsberg, K. P., Farhadifar, R., Ranft, J., Umetsu, D., Widmann, T. J., Bittig, T., et al. (2009). Increased cell bond tension governs cell sorting at the *Drosophila* anteroposterior compartment boundary. Current Biology: CB, 19(22), 1950–1955. http://doi.org/10.1016/j.cub.2009.10.021).

2) In Figure 6, for clarity I would add a drawing of the corresponding segments when 1 cell and 2 cells are missing in the supplementary information.

This is a good idea. We have added two example scenarios in Figure 6—figure supplement 1, where cells from stripe 3 and cells from stripes 3 and 4 are missing, respectively. We have adjusted the legends and text accordingly.

3) In Figure 2, The Figure 2 could be placed in supplementary information and the Figure 2 scale bars are missing as well as a scale for the time.

We have added scale bars to Figure 2 and time-scale to Figure 2.

Figure 2 could be relegated, or reduced to an inset of Figure 2, but this would not save space, given the 4x3 layout, without changing the look of the whole figure. We would prefer to leave Figure 2 as is.